# Comparison of total water vapour content in the Arctic derived from GNSS, AIRS, MODIS and SCIAMACHY

Dunya Alraddawi[1], Alain Sarkissian[1], Philippe Keckhut[1], Olivier Bock[2], Stefan Noël[3], Slimane Bekki[1], Abdenour Irbah[1], Mustapha Meftah[1], Chantal Claud[1,4]

[1]OVSQ-LATMOS, University Paris Saclay, Guyancourt 78280, France
[2]IGN-LAREG, University Paris Diderot, Paris 75013, France
[3]Institute of Environmental Physics, University of Bremen, 28334 Bremen, Germany
[4]LMD/IPSL, CNRS, École Polytechnique, Université Paris Saclay, ENS, PSL Research University, Sorbonne Universités, UPMC Univ Paris 06, Palaiseau, France

*Corresponding author*: Alain Sarkissian (alain.sarkissian@latmos.ipsl.fr)

**Abstract.** Atmospheric water vapour plays a key role in the Arctic radiation budget, hydrological cycle and hence climate, but its measurement with high accuracy remains an important challenge. Total Column Water Vapour (TCWV) data set derived from ground-based GNSS measurements are used to assess the quality of different existing satellite TCWV datasets, namely from the Moderate Resolution Imaging Spectroradiometer (MODIS), the Atmospheric Infrared Sounder (AIRS), and the SCanning Imaging Absorption spectroMeter for Atmospheric CHartographY (SCIAMACHY). The comparisons between GNSS and satellite data are carried out for three reference Arctic observation sites (Sodankyla, Ny-Alesund and Thule) where long homogeneous GNSS time series of more than a decade (2001-2014) are available. We select hourly GNSS data that are coincident with overpasses of the different satellites over the 3 sites and then average them into monthly means that are compared with monthly mean satellite products for different seasons. The agreement between GNSS and satellite time series is generally within 5% at all sites for most conditions. The weakest correlations are found during summer. Among all the satellite data, AIRS shows the best agreement with GNSS time series, though AIRS TCWV is often slightly too high in drier atmospheres (i.e. high latitude stations during fall and winter). SCIAMACHY TCWV data are generally drier than GNSS measurements at all the stations during the summer. This study suggests that these biases are associated with cloud cover, especially at Ny-Alesund and Thule. The dry biases of MODIS and SCIAMACHY observations are most pronounced at Sodankyla during the snow season (from October to March). Regarding SCIAMACHY, this bias is possibly linked to the fact that the SCIAMACHY TCWV retrieval does not take accurately into account the variations in surface albedo, notably in the presence of snow with a nearby canopy as in Sodankyla. The MODIS bias at Sodankyla is found to be correlated with cloud cover fraction and is also expected to be affected by other atmospheric or surface albedo changes linked for instance to the presence of forests or anthropogenic emissions. Overall, the results point out that a better estimation of seasonally-dependent surface albedo and a better consideration of vertically-resolved cloud cover are recommended if biases in satellite measurements are to be reduced in the polar regions.

## 1 Introduction

Water vapour has an important role in the Earth radiative balance (e.g. *Kiehl and Trenberth, 1997; Trenberth and Stepaniak, 2003; Ruckstuhl et al., 2007; Trenberth et al., 2007)*, hydrologic cycle (e.g. Chahine, 1992; Serreze *et al.*, 2006; Jones *et al.*, 2007; Hanesiak *et al.*, 2010); and climat change (e.g. *Schneider et al., 1999, 2010; Held and Soden, 2000; Ramanathan and Inamdar, 2006; Rangwala et al., 2009)*. The rate of the Arctic climate change is twice larger than the global one due to greenhouse gases (GHG) increase. The water vapour feedback loop is highlighted, as part of many others, responsible of the Arctic amplification (e.g. *Winton, 2006; Francis and Hunter, 2007; Miller et al., 2007; Screen and Simmonds, 2010; Chen et al., 2011; Ghatak and Miller, 2013)*.

Water vapour measurements (total column / vertical profile information) are available using radiosondes since early 1940s and satellites since 1980s primarily for meteorological purposes, while Global Positioning System (GPS) and more generally Global Navigation Satellite System (GNSS) measurements, have been diverted from positioning to remote sensing of atmospheric water vapour since the1990s *(Bevis et al., 1992)* .

The Total Column of Water Vapour (TCWV), also called Integrated Water Vapour (IWV), is defined as the density of water vapour in an atmospheric column over a unit area (kg m$^{-2}$). It is also sometimes referred as Precipitable Water (PW), which represents the height of liquid water (in mm) resulting from the condensation of all the water vapour of a vertical column over a unit area.

TCWV is characterized by large spatial and temporal variability. It affects the water cycle intensity and the atmospheric dynamics *(Sherwood et al., 2010; Trenberth et al., 2005)*. Since 2010, the Global Climate Observing System (GCOS) declared the TCWV as an essential climate variable, and highlighted the importance of high resolution long time series that could enable the detection of both local and global TCWV trends.

The available satellite remote sensing techniques to observe TCWV in Micro Wave (MW), Infra Red (IR), Near Infra Red (NIR), and VISible (VIS) spectral domains are promising, with a global coverage that enables climate studies, but with limited retrieval capability (e.g. only day time, only clear skies, or over oceans only). Satellite observations are validated by ground-based techniques, traditionally radiosondes. However, radiosonde data suffer sometimes from systematic observational errors, and spatial and temporal inhomogeneity and instability *(Gaffen, 1994; Wang, 2003)*, that could induce potentially regional biases, if radiosondes alone are used to validate satellite data (*Wang and Zhang, 2008, 2009; Bock and Nuret, 2009)*.

GNSS measurements complete the global radiosonde observations  as another reliable reference to validate satellite water vapour retrievals and atmosphericl models (e.g. *Bock et al., 2007*, and references therein). GNSS TCWV measurements are independent of the weather, performed with high temporal resolutions (a few minutes), and have continuously improved spatial resolution (from global down to a few km for local networks). While GNSS is based on a delay measurement, it can be applied similarly to different sensors, and is an ideal tool for long-term measurements, despite the presence of a possible bias in certain specific configurations *(Ning et al., 2016)*.

Many studies comparing global satellite TCWV products with radiosonde, GPS, and other reference data have pointed a dependence in bias and RMSE to various observational factors like TCWV content (larger biases and RMSE are generally observed in regions with higher TCWV), reduced extreme values (e.g. wet bias at low TCWV values and dry bias at large values), solar zenith angle dependence (increased radiative transfer model error with

5 larger zenith angles), day/night difference (increased background noise at daytime for VIS and NIR techniques), seasonal dependence (related to the two previous factors), latitude/geographical dependence (also partly connected with the former), and cloudiness dependence (usually increased biases and scatter with increasing cloudiness). Many of these aspects are discussed by *(Vaquero-Martínez et al., 2017)* for VIS, NIR, and IR techniques over the Iberian Peninsula. Few studies investigated the polar and snow-covered regions. For example, *(Thomas et al., 2011)*

10 compared GPS to MODIS and AIRS over 13 Antarctic stations for 2004, and found that GPS TCWV data are drier than MODIS, while wetter than AIRS. *(Palm et al., 2010)* compared GPS with SCIAMACHY and GOME-2A data over Ny-Alesund/Arctic, and found GPS to under-estimate both satellite sensors.

The current study provides inter-comparisons of various measurements and methods allowing to quantify uncertainties, accuracies, and limitations of several global satellite sensors/techniques available.

15 As common reference, we use a recently reprocessed version of GPS TCWV data with hourly temporal sampling covering the period from 1996 to 2014. It enables the largest number of coincident overpasses of three independent selected satellites AIRS/IR (from 2003 to 2014), MODIS/NIR (from 2001 to 2014), and SCIAMACHY/VIS (from 2003 to 2011) for inter-comparisons. Three Arctic ground-based GNSS observation sites were chosen: Ny-Alesund (78°N, 12°E), Thule (76°N, 69°W), and Sodankyla (67°N, 26°E). Satellite gridded data were matched with these

20 stations within a maximum spatial distance of 50 km.

Generally, satellites measurements are more accurate during clear sky conditions. In this work we use only cloud cleared products in order to assess their uncertainties in optimal conditions in the Arctic region. However, cloud clearing is a challenging task. For this reason, we investigate the possible relation between satellite TCWV biases and the cloud cover at various time scales (seasonal and inter-annual, using time series with monthly, seasonal and

25 annual sampling). In order to strengthen the conclusions, two different cloud fraction products are used (from MODIS and AIRS measurements). Though cloudiness dependence is not the only error source in satellite TCWV retrievals, it is one of the least well known, especially for the Arctic region. The impact of clouds on TCWV retrievals is to shield partly or totally, depending on the cloud opacity, the underlying atmosphere, so that the observed radiance is only a measure of the water vapor content above the cloud. The mixing of cloudy pixels with clear pixels tends to lower the TCWV estimate and lead to a dry bias. On the other hand, depending on wavelengths,

30 multiple scattering inside the clouds may increase the observed radiance and lead to over-estimation of the water vapor content above the cloud. These effects are usually corrected in the retrieval algorithms using different methods depending on the instrument. However, in the end, both under- and over-estimation of the retrieved TCWV can be observed.

35 Section 2 describes the datasets used and discussed the error sources specific to each technique. Section 3 presents results of TCWV comparisons (satellite retrievals compared to GNSS). Section 4 investigates the link between observed biases in the satellite data and cloudiness. Section 5 presents conclusions.

## 2 Description of the data sets

### 2.1 GNSS

Originally designed for real-time navigation and positioning, GNSS was rapidly seen as a cheap and accurate technique for measuring TCWV from the ground (Bevis et al., 1992). The principle consists in estimating the propagation delay induced by the atmosphere of the microwave signals emitted by the GNSS satellites and received by ground-based receivers. The Zenith Tropospheric Delay (ZTD) is usually parsed into its wet and hydrostatic components (ZWD and ZHD, respectively for Zenith Wet Delay, and Zenith Hydrostatic Delay). Accurate estimations of surface pressure and a weighted mean temperature are required to convert GNSS ZTD into TCWV using the following formulas (Bevis et al., 1992) :

$$ZWD = ZTD – ZHD, \tag{1}$$

where ZTD is the GNSS ZTD estimate, ZHD is computed from the surface pressure *(Davis et al., 1985)*:

$$ZHD = 0.002277 \, P_{sfc} / f \, (\lambda, H),$$

Where $P_{sfc}$ is the surface pressure, $\lambda$ and $H$ are the latitude and altitude of the station, $f (\lambda, H)$ accounts for the ˇ geographical variation of the mean acceleration due to gravity *(Davis et al., 1985)*.

TCWV is converted from the ZWD as:

$$TCWV = ZWD * K \, (T_m), \tag{2}$$

Where $K (T_m)$ is a delay to mass conversion factor and $T_m$ is the humidity-weighted mean temperature (Bevis et al., 1992).

In this study, we used GNSS ZTD data from the Geodetic Observatory Pecny (Czech Republic) named "repro2 solution" and referred to as GO4 *(Dousa et al., 2017)*. This ZTD dataset was produced with a homogeneous and optimized processing of GPS observations. Outliers in the ZTD time series were detected and removed using the range-check and outlier check method described in *(Bock et al., 2014)*. ZHD and $T_m$ were computed from the ERA-Interim reanalysis pressure level data (37 vertical levels between 1000 hPa and 1 hPa, 0.75° x 0.75° horizontal resolution, 6-hourly time resolution) *(Dee et al., 2011)*. The data were first interpolated vertically to the height of the GNSS station and then interpolated horizontally (bi-linear interpolation using the 4 grid-points surrounding the station) to the location of the station. The 6-hourly $P_{sfc}$ and $T_m$ data were then interpolated (with cubic splines) to the times of the GNSS ZTD data resulting in the final 1-hourly GNSS TCWV dataset.

In order to overcome the satellite/GNSS timing error due to limited hours of MODIS/AIRS/SCIAMACHY measurements during a month over a fixed point at the surface, the satellites passing hours over the three Arctic GNSS stations were defined through the IXION software (http://climserv.ipsl.polytechnique.fr/ixion/index.php). For each satellite, only GNSS TCWV corresponding to the over-passes less than 1 hour (Table 1) were used to calculate the corresponding monthly time series.

Seasonal variations of the TCWV over all three sites for a common period of 11 years (2004-2014) exhibit a pronounced seasonal cycle (Fig. 1) with mean values ranging from a maximum in July of 20, 14, 13 kg m$^{-2}$, to a minimum in winter of 6, 4.5, 2 kg m$^{-2}$ over Sodankyla, Ny-Alesund and Thule respectively.

Extreme hourly values could reach 40 kg m$^{-2}$ (not shown) over Sodankyla. This highest amplitude appears in summer under continental climate conditions. Ny-Alesund and Thule have likely similar seasonal features. However,

Thule has drier winter/fall periods due to the Greenland ice sheet climate effect. "Figure 2 shows that the year to year variations of TCWV at the three stations are smaller than the seasonal cycle (Fig.1). This can be easily seen for summer values (peak values)."

## 2.2 MODIS

The Moderate resolution imaging spectroradiometer (MODIS) is installed on both platforms (Terra and Aqua) of the

Earth Observing System (EOS). Both satellites are launched on polar orbits since 1999 (Terra) and 2002 (Aqua). They overpass the equator at 10:30 a.m. and 1:30 p.m., respectively. The global coverage is provided within 1-2 days, through a nadir-looking geometry at a solar zenith angle of 45 degrees. The spatial resolution varies between 250 m and 1 km per pixel depending on the spectral band.

MODIS observes the NIR solar radiation reflected by sufficiently bright surfaces and clouds and IR thermal emission

in 36 channels covering the spectral region 0.4 - 14.4 μm. It allows the measurement of many other trace gases in addition to clouds and aerosols. In this study we use only the NIR data as they are known to be more accurate.

Five NIR channels are used for retrieving daytime water vapour. They are centred on 0.865, 0.905, 0.936, 0.94, and 1.240 μm, in which all the surface types are sufficiently bright (albedo > 0.1). The extreme channels (0.865 and 1.240 μm) have no water vapour absorption features. They are used to estimate the surface reflectance. The three

other channels (0.905, 0.936, and 0.94 μm) absorb water vapour with different sensitivity. The 0.936 μm channel has the strongest absorption sensitivity. TCWV is derived by a differential absorption technique involving channels with absorption and channels without. The accuracy of this product is claimed to be 5–10% *(Gao and Kaufman, 2003)*. Main uncertainties concern the spectral reflectance of surface targets and the uncertainty in the amount of haze for dark surfaces under typical atmospheric conditions *(Gao and Kaufman, 2003)*.


The TCWV data used in this study are from the version 6 of the MODIS instrument on board Terra platform, referenced as "Water vapour near infrared - clear column (bright land and ocean sunglint only): Mean of Daily Mean" *(Gao and Kaufman, 2003; Hubanks et al., 2008)*. The Aqua platform was not used because of many gaps in the measurement. We retrieved global monthly mean files, gridded at 1° by 1°, from the MOD08_M3.006 data

stream[1] freely available on:

ftp://ladsweb.nascom.nasa.gov/allData/6/MOD08_M3.

TCWV we extracted from 2001 to 2014 for Sodankyla and Ny-Alesund and from 2004 to 2014 for Thule for the comparison with GPS. The pixel selection method is the following. MODIS data coordinates refer to the centre of

---

[1] Dataset DOI: http://dx.doi.org/10.5067/MODIS/MOD08_M3.006

each gridded pixel, so a single pixel is considered per station (to avoid interpolation and select the nearest pixel to GNSS/IGS stations) and defined as follow:

$$(Lat, Lon)_{(Pixel)} = (lat, lon)_{(station)} + (0.5°, 0.5°),\ \ \ \ \ \ \ \ \ \ \ \ \ \ \ (3)$$

Where $(lat, lon)_{station}$ are defined in Table 1 for each of the three stations.

For example the Sodankyla MODIS pixel was selected as follow:

$(Lat, Lon)_{(Soda)} = (67°, 26°)_{(table1)} + (0.5°, 0.5°) = (67.5°, 26.5°)$

MODIS cloud fraction *(Hubanks et al., 2008; Platnick et al., 2003)* taken from the same atmospheric product (MOD08_M3.006) is used also to test the sensitivity of the satellite measurements to the presence of clouds. This product is thought to be efficiently capable to detect low-level clouds in dry atmospheres (Ackerman et al., 2008). MODIS cloud fraction is defined as the ratio of the count of the lowest two clear sky confidence levels (cloudy & 
probably cloudy) to the total count of scenes per 1° *1°.

**2.3 SCIAMACHY**

Launched on board the satellite ENVISAT-1in March 2002, the Scanning Imaging Absorption spectrometer for Atmospheric CHartographY (SCIAMACHY) was designed to observe the earthshine radiance and the solar irradiance within limb and nadir alternating viewing geometry. SCIAMACHY nadir and limb observations cover the 
spectra from Ultra Violet (UV) to NIR (214-2380 nm) at moderate spectral resolution (0.2-1.5 nm). The observed spectra enable the measurement of many other trace gases, as well as clouds and aerosols.

SCIAMACHY can measure water vapour at various wavelengths from the VIS to the SWIR (Short-Wave Infrared). This paper uses TCWV retrieved by the Air Mass Corrected Differential Optical Absorption Spectroscopy method, shortly AMC-DOAS *(Noël et al., 2004),* where water vapour is measured in nadir mode in the visible part of the 
spectrum between 688 nm and 700 nm. This method makes use of the similar slant optical depth of both $O_2$ and water vapour to determine an Air Mass correction Factor (AMF) which compensates for insufficient knowledge of the atmospheric and topographic background, like surface elevation and clouds. AMF includes a correction for the part of the atmosphere below the cloud, but this relies on some assumptions (e.g. about profile shapes) which might lead to under- or over-correction of TCWV values.

Though SCIAMACHY TCWV measurements are independent of the initial humidity profile, they are affected by other factors. A dominant error source in SCIAMACHY TCWV retrieval is caused by uncertainties of the atmospheric radiative transfer, mainly due to effects of varying cloud cover and surface albedo for different surfaces *(Wagner et al., 2011)* This error source is estimated to be about 15 % for clear sky observations, and up to 100% in large clouds amounts (*Van Malderen et al., 2014*). The sensitivity to the surface albedo may cause deviations of up 
to about 15%, or 6 kg m$^{-2}$ in regions of high surface albedo *(Noël, 2007a; Noël et al., 2004)*. A scatter of about 5 kg m$^{-2}$, caused by atmospheric variability, is usually observed during the inter-comparison with other TCWV datasets *(Noël, 2007b),*

The three stations used in this study were part of the ground-based stations contributing to the SCIAMACHY validation effort *(Piters et al., 2006)* during which water vapour profiles alone were validated over Thule and 
Sodankyla, while TCWV was additionally validated over Ny-Alesund.

TCWV data used in this paper are from *(Noël et al., 2004)*, where all observations with AMF < 0.8 were removed, as well as those performed at solar zenith angles larger than 88°. We apply an extra screening that excludes data with SCIAMACHY indicated error > 20 % (fitting error), and swath data of spatial distance more than 50 km (actually 54 km) to the station coordinates defined by Table1.

This collocation is made by choosing data that meet the condition:

$| \text{Lat}_{(data)} - \text{lat}_{(station)} | \leq 0.5°$   and      $| \text{Lon}_{(data)} - \text{lon}_{(station)} | \leq 0.2°,$                    (4)

This surface is defined according to SCIAMACHY swath data footprints size which is about 30 km × 60 km.

Then, SCIAMACHY TCWV monthly means are calculated from all the matched data to the given station. Note that SCIAMACHY data solar dependency results in missing data for winter months. Our study takes place from 2003 to 2011 over Sodankyla and Ny-Alesund and from 2004 to 2011 for Thule.

**2.4 AIRS**

The Atmospheric Infrared Sounder (AIRS) is carried on Aqua/EOS since May 2002. This platform has an equatorial over passing at 1:30 p.m. with a sun-synchronous orbit. AIRS was dedicated to water cycle, energy, and traces gases observations. It provides twice daily global coverage with higher vertical resolutions than all previous sensors, and comparable accuracy to radiosondes *(Tobin et al., 2006)*. AIRS is a hyper-spectral scanning infrared sounder. It

measures upwelling thermal radiation emitted from the atmosphere and the surface. However, almost 30% of the AIRS radiances could be trapped below clouds *(Susskind et al., 2006)*. These possible profiles could be better retrieved using simultaneous observations from the Advanced Microwave Sounding Unit (AMSU) *(Lambrigtsen, 1999)* in a process called "cloud-clearing" *(Susskind et al., 2003)*. The observation geometry of these combined measurements or the AIRS Field of Regard (FOR) is called "AIRS golf ball".

Humidity profiles (level 2 products) are retrieved from cloud-cleared radiances (level 1). A set of different water vapour sensitive channels are used in addition to temperature sensitive channels. Water vapour mixing ratios at certain pressure levels are retrieved using the Radiative Transfer Algorithm AIRS-RTA described by *(Strow et al., 2003)*. TCWV is obtained by integrating the vertical profile of water vapour mixing ratio.

The RMSE of the AIRS water vapour profiles is estimated to 10-15% over 2-km layers in the troposphere (Divakarla et al., 2006; Fetzer et al., 2003). Several studies have confirmed that both the AIRS radiances and the AIRS clear-sky forward model have an absolute accuracy of around 0.2 K for the spectral channels used in temperature and water vapour retrievals *(Fetzer et al., 2003; Strow et al., 2006)*. AIRS  TCWV retrievals are mainly limited by the accurate initialization of the humidity profile *(Fetzer et al., 2006)*.

Previous versions of AIRS TCWV were validated against radiosondes over oceanic areas *(Fetzer et al., 2006),* and against reanalysis (ECMWF) *(Susskind et al., 2006)*. *Gettelman et al. (2006)* showed that AIRS retrievals in polar regions are unbiased relative to in-situ radiosondes. Most results indicate a small mean bias that doesn't exceed 10 % with no significant dependency upon cloud amount.

AIRS TCWV data[3]  used in this study is from the version 6, monthly weighted means, level 3 product, referenced as AIRX3STM**.006**. It presents a standard physical retrieval that includes both AIRS and AMSU radiances *(Susskind et al., 2014)*. This data set has dense orbital coverage at high latitude. Similarly to MODIS data, the 1° by 1° gridded AIRS pixels were screened. The AIRS considered TCWV pixel per station is the same as for MODIS and defined by formula (3). The comparison to GNSS is done from 2003 to 2014 for Sodankyla and Ny-Alesund and from 2004 to

2014 for Thule according to AIRS and GNSS data availability.

During this study, we additionally use the AIRS effective cloud fraction *(Kahn et al., 2014)* monthly 1° by 1° data set from the same atmospheric product (AIRX3STM.006) in order to investigate possible effects of cloud interference on the satellites observed biases. The AIRS effective Cloud fraction product is the multiplication of spatial cloud fraction and cloud emissivity. AIRS cloud fraction is defined as the ratio of the number of AIRS cloudy

measurements (CF>0.01) to the total number of AIRS measurements per 1° by 1°.

### 3 Mean seasonal comparisons and discussion

### 3.1 GNSS vs MODIS

MODIS time series of monthly means TCWV are compared to monthly means of coincident overpassing (mentioned in Table 1) GNSS data over Sodankyla and Ny-Alesund for the period 2001-2014, and over Thule for 2004-2014.

This difference in the data range is linked to the GNSS data availability, as GNSS dataset has some missing values at Thule during 2001-2003. The results show an excellent overall agreement with a high coefficient of correlation R > 96 % for the monthly time series (Table 2). High correlation of the monthly time series is indeed expected since the seasonal cycle is very marked at all three sites (Fig. 2). The mean biases are +0.4, +0.6, +1.7 kg m$^{-2}$ at Ny-Alesund, Thule, and Sodankyla, respectively (Table 2). The overall positive biases indicate that MODIS generally under-

estimates TCWV compared to GPS. This was previously reported over other cold regions of the world, using other versions of GNSS and MODIS data, for example, over the Tibetan plateau for both stations Gaize and Naque *(Liu et al., 2006)*. Here we can also notice a latitudinal decrease both in the absolute bias (in kg m$^{-2}$) and the relative bias, as well as in the root mean square errors (RMSE), which means that the TCWV retrieval is actually more accurate at higher latitudes.

The mean biases and inter-annual variability of the individual months are analysed with boxplots in Fig. 3. A seasonal variation can be seen at all three sites in the bias and in the dispersion (see the inter-quartile range in the boxplots). The largest variations are observed at Sodankyla with large positive biases between September and February, and slightly negative biases between July and August.

Dividing the year into four seasons, the statistics were also calculated and given in Table 2. At Ny-Alesund and

Thule the relative bias doesn't exceed 13% regardless of the season and the absolute biases are larger in (June-July-August) JJA and SON (September-October-November). A small wet bias is observed at Ny-Alesund during spring which was also reported for Antarctica during the transition seasons *(Thomas et al., 2011)*. The inter-annual variability is best represented for the DJF (December-January-February) and SON seasons at both high latitude sites

---

[3] https://disc.gsfc.nasa.gov/datasets/AIRX3STM_V006/summary?keywords=airs%20version%206

(Ny-Alesund and Thule) with correlations in the range 56 – 83% (all significant) but quite poorly in JJA with correlation values of 10 and 15% (not significant). The larger biases and lower correlations in JJA are linked with cloud cover (see section 4).

At Sodankyla, the results are more complex to interpret. Multi factors are involved with the observed biases including clouds. During the snow season which lasts from October to April at Sodankyla, the solar angle has a

strong influence on the effective albedo, since Sodankyla is totally covered with canopy, unlike both other stations, and its forests intercept the majority of incoming solar radiation, as pointed out by *(Gryning et al., 2002)*. Additionally, Sodankyla snow samples contain higher impurity concentrations (black carbon) than measured elsewhere in Arctic Scandinavia or Greenland *(Doherty et al., 2010),* as well as a bigger snow grain size. These two factors contribute to a decrease in surface albedo *(Meinander et al., 2013)*. The chemical exchange between polluted

atmospheric layers due to winter biomass burning and snow surface opaque the lower part of the atmosphere at the instrument's wavelengths. Since the MODIS retrieval capacities are sensitive to surface albedo and atmospheric transmittance (Section 2.2), the seasonal variation in these parameters and could explain the variation in the MODIS TCWV bias, especially the dry bias during the snow season at Sodankyla. During summer at Sodankyla, MODIS TCWV estimates were found higher than GNSS TCWV measurements. This opposite bias can be explained by the

fact that the snow coverage nearly disappeared, in addition to the tendency of increasing MODIS TCWV with increasing water vapour at sites below 3000 m *(Lu et al., 2011)*. This bias is also found to be correlated with MODIS cloud fraction (Section 4).

**3.2 GNSS vs SCIAMACHY**

Calculated monthly means of SCIAMACHY TCWV over Sodankyla and Ny-Alesund for 2003-2011 and over Thule

for 2004-2011 were compared to means of coincident GNSS measurements. This comparison doesn't include winter pairs over Thule and Ny-Alesund because of missing SCIAMACHY measurements during polar winter. Similarly to MODIS, SCIAMACHY under-estimates TCWVs at all three sites with mean absolute biases between 0.6 and 2.4 kg m$^{-2}$ and relative biases between 6 and 22% (Table 2, monthly mean biases). The dry bias agrees well with previous findings at high-latitude sites by *(Van Malderen et al., 2014)* using different versions/retrieval methods of both

GNSS and SCIAMACHY data (namely their SCIAMACHY data were completed with TCWV data from other satellites to achieve higher time sampling and temporal coverage). A good overall correlation is observed between SCIAMACHY and GNSS monthly time series with R>90 % and RMSE between 24 and 27%. The monthly mean biases (Fig. 4) show also a marked seasonal variation at all three sites. The absolute biases show a similar seasonality at all stations, having their minimum during spring and maximum during summer or fall. At Ny-Alesund and Thule,

the dry biases are the largest during SON and JJA, similar to MODIS but with different magnitudes. At Sodankyla the bias is around 5 kg m$^{-2}$ in JJA, i.e. much larger and of opposite sign compared to MODIS (Table 2). The seasonal RMSE values are generally larger as well compared to MODIS at Ny-Alesund and Thule but smaller at Sodankyla where they don't exceed 30%. Inter-annual variability is generally well represented by SCIAMACHY at Ny-Alesund and Thule (R > 76% significant in all seasons except at Thule in JJA). At Sodankyla the correlations are much

smaller, similarly to what we found with MODIS.

Consideration of surface albedo of complex surfaces could be also a challenge for the SCIAMACHY TCWV retrieval. The presence of snow with a nearby canopy (e.g. in Sodankyla) might result in a surface albedo significantly different from the prescribed surface albedo used in the AMC-DOAS method (e.g. 0.05 compared to 0.5) which would explain the winter biases (Noël, 2007b). Nevertheless, the DJF and SON absolute TCWV biases found here with SCIAMACHY are smaller than those found with MODIS. They are also smaller than those expected

for SCIAMACHY in such conditions (Noël, 2007b). On the other hand, the JJA bias at Sodankyla is the most challenging and yet unexplained issue.

**3.3 GNSS vs AIRS**

The AIRS TCWV monthly product shows excellent agreement with coincident GNSS measurements at all stations. The overall correlation with GNSS is larger than 98 %, and the mean bias is smaller than 1 kg m$^{-2}$ in absolute value

(Table 2). These biases are in the same range as reported in previous studies over cold regions, e.g. *Thomas et al. (2011)* over Antarctica. However, our study uses a more recent and improved version of both AIRS and GNSS data sets. Again, the monthly mean biases show a distinct seasonal variation at all three sites (Fig. 5). AIRS is found to be biased wet compared to GNSS during the colder and drier periods and biased dry during the moister months over Ny-Alesund and Thule (Fig. 5). This observed wet/dry seasonal variation of the bias is consistent with the previous

validation efforts of *Rama Varma Raja et al. (2008)* and of *Van Malderen et al. (2014)*. The bias at Sodankyla follows similar seasonal variation but with an overall offset (the bias is always positive). The inter-annual variability is globally much better reproduced by AIRS than the two previous sensors as attested by the correlation coefficients > 64% (all significant except one). The correlations are higher over Ny-Alesund and Thule than Sodankyla. Compared to MODIS and SCIAMACHY, the results are noticeably better at Sodankyla (seasonal bias and RMSE <

13% and 17%, respectively). So there must be a significantly different sensitivity in the measurements to the atmospheric properties over Sodankyla. In the next section we investigate more specifically the impact of cloud cover on the TCWV retrievals from all three sensors.

**4 Cloud impact on TCWV observations**

MODIS and SCIAMACHY TCWV measurements are known to be sensitive to the presence of clouds, whereas the

AIRS TCWV product is less impacted by clouds as it includes microwave water vapour measurements and a robust cloud clearing technique also based on microwave measurements *(Susskind et al., 2003)*. This section uses the AIRS and MODIS cloud fraction products to examine the correlations between the TCWV biases found in section 3 and cloud cover. The use of both products helps to minimise the influence of different overpasses between clouds fraction and satellites measurements. In this study, AIRS and MODIS cloud fractions show similar annual cycles

only at Thule. This is not surprising, previous comparisons between both cloud fractions showed the largest disagreement over the high latitudes (e.g. (Wu et al., 2009)). The observed inconsistencies in both cloud fractions are expected to be dominated by retrieval algorithm differences instead of differences in the observed radiances (Kahn et al., 2007). More significant differences between AIRS and MODIS retrievals can be found in areas of low clouds in the Arctic in summer (Weisz et al., 2007) as AIRS is less capable to detect the multi layers summer clouds.

However, AIRS is better suited to retrieve thin cirrus than MODIS (e.g.(Kahn et al., 2007); (Weisz et al., 2007)) especially during the polar winter and at night time (Kahn et al., 2005). Additionally, AIRS retrieval of Cloud Top Pressure performs better than MODIS retrievals over polar regions , especially in presence of low-level temperature inversions (Weisz et al., 2007).

    Figure 6 describes the annual cycle of cloud fraction at the three sites based on monthly mean AIRS cloud fraction

product for a common period of 11 years (2004-2014). At Sodankyla, the 8-months period from May to December shows a cloud cover above 50%, with a maximum in June (> 60%) and a minimum in March (< 40 %). At Thule, the seasonal variation is even larger with 4 months < 35% (January to April) and 4 months > 50%. September has the cloudiest conditions (> 60 %) and April has the clearest (< 30%). At Ny-Alesund, cloud cover is above 44% all year long, with values > 50% during 9 months and a relative minimum (<50%) during the JJA summer months. In this

section we examine the correlation coefficients between monthly TCWV biases and cloud cover with different temporal sampling. We start with the full time series of monthly means , then move on to the annual cycle (averages over all years for each of the 12 calendar months), next the inter-annual variability cycle by calendar season (DJF, MAM, JJA, SON) and finally the inter-annual variability by month. Table 3 illustrates the correlations of TCWV biases with AIRS cloud fraction (AIRS CF), and Table 4 with MODIS (MODIS CF).

## 20   4.1 GNSS vs MODIS

    Although this study uses only clear column water vapour observations, the monthly time series of TCWV differences (GNSS-MODIS) show significant correlations with the coincident cloud fraction AIRS (MODIS) at Thule and Ny-Alesund, with R = 39(44) and 44(19) % respectively (all significant values are given at a significance level > 95%), at Sodankyla, a significant correlation of 49% is found only with MODIS cloud fraction (Table 3,4) which means

that the results at this station are more affected by the different overpasses of AIRS cloud fraction.

    The annual cycle of TCWV biases shows significant correlation with coincident cloud fraction (both MODIS and AIRS) at Thule with R = 68 %. Unlike Ny-Alesund, this different sensitivity is due to the stronger annual cycle of both cloud fractions at Thule in comparison to Ny-Alesund (see AIRS CF in Fig. 6). Again, at Sodankyla, the correlation is observed only with MODIS cloud fraction (R= 70%).

The inter-annual variability is generally more dominant at Ny-Alesund out of winter (R = 58% in JJA and MAM (AIRS CF), R= 58% in SON (MODIS CF)), both cloud fractions are significantly correlated with TCWV biases in August and September, while only AIRS (MODIS) cloud fraction have 7(4) significant months with R > 58 %.

    At Thule, the inter-annual variability is significant with both cloud fractions on summer, and additionally on winter and fall when AIRS CF is used (Table 3). At monthly scale, only two months are significant with AIRS CF

(November and December with R = 77, 70% respectively), while two other months are significant with MODIS CF (July and August with R = 81, 66%).

    The high correlations between TCWV biases and cloud cover in JJA at both sites could explain the poor agreement found in section 3.1 (large biases 0.6 and 1.1 kg m$^{-2}$, and small correlations R = 10 and 15%, see Table 2) between MODIS and GNSS TCWV time series at Ny-Alesund and Thule respectively. Figure 7 gives more insight into the

time series at all sites in summer.

Regarding Sodankyla, TCWV differences show significant correlation with both cloud fractions on November with R> 58%, while only with coincident MODIS cloud fraction at monthly(R = 49%), annual (R= 70%), and summer inter-annual variability (R= 56%). TCWV biases are correlated with AIRS CF during some months of the snow season (R = 76 and 84 % in December and March respectively), while with MODIS CF during May (R= 77%). Consequently, cloud cover may contribute to part of the dry biases in DJF, SON, and JJA reported at Sodankyla in

section 3.1. However, biases at this site are probably not dominated only by cloud effects. We believe that the environmental features of Sodankyla, which complicate the surface albedo estimation, are also responsible of limiting MODIS retrieval capabilities as previously discussed in section 3.1.

## 4.2 GNSS vs SCIAMACHY

The monthly time series of SCIAMACHY TCWV biases are significantly correlated at Thule with R>58% for both

cloud fractions, and on the annual scale with R>75%.

At Ny-Alesund and Sodankyla, monthly TCWV biases show similar sensitivity to AIRS cloud fraction only, with R = 26%, 29% respectively.

The correlations at annual scale at Thule and Ny-Alesund behave again like in 4.1. They increase at Thule (from R = 60 % (AIRS CF), 58 % (MODIS CF) at monthly scale to R = 75% (AIRS CF), 80 % (MODIS CF) at annual scale)

and decrease at Ny-Alesund (from R = 26% to -19% (AIRS CF only)), while at Sodankyla the annual variations are strongly correlated (only with AIRS CF) at R = 75 %.

SCIAMACHY's TCWV retrieval is more sensitive to cloud cover than MODIS when AIRS CF is used, but MODIS retrieval shows more sensitivity to MODIS CF. Different sensitivity is observed to each of the used cloud fraction products, which is probably linked to closer SCIAMACHY overpasses with AIRS CF than with MODIS CF. The

results at Sodankyla are thought to be more influenced by the diurnal variability, and thereby the matched passing hours (CF & satellites). Similar sensitivities to both cloud fractions are marked in red in Table 4.

Generally, our results agree with the findings of Palm et al., (2010) who concluded that cloudy conditions introduce a severe bias at Ny-Alesund, even if the SCIAMACHY measurement passes the cloud screening filter.

As found with MODIS (section 4.1), TCWV biases and both cloud fractions are strongly correlated at the inter-

annual scale in JJA at Thule with R >69% (Table 4).

At Ny-Alesund, TCWV biases are correlated with both cloud fractions in August, while only with AIRS CF for the whole summer with R = 72 % (Table 3). Figure 8 shows the strong inter-annual variability in JJA at all sites.

At Sodankyla, the inter-annual variability in TCWV biases and both cloud fractions is significantly correlated in May with R = -77%, -56% (AIRS CF and MODIS CF respectively). This anti-correlation is not explained yet.

## 4.3 GNSS vs AIRS

The results with AIRS are quite different compared to SCIAMACHY and MODIS. Whereas monthly time series of TCWV biases show significant positive correlation with both cloud fractions at Thule (R = 31% (AIRS CF), 41% (MODIS CF)) similar to SCIAMACHY and MODIS (Table 4). The correlation is negative or absent at Ny-Alesund (R = -42% (AIRS CF), 4% (MODIS CF)). The negative correlation at Ny-Alesund is explained by the pronounced

but opposite annual variations of the TCWV biases (Fig. 5) and the AIRS cloud cover (Fig. 6) at this site, with an annual correlation of R = -94% (Table. 3).

Overall, Ny-Alesund TCWV AIRS biases seasonality is almost linear with negative slope with AIRS cloud fraction. Moreover, The inter-annual variability of TCWV biases and AIRS cloud cover is significant at Ny-Alesund only for DJF (R = -63%) and a few individual months (Table 3). The dominated wet biases in winter (AIRS measurements are bigger than those of GNSS unlike SCIAMACHY and MODIS, see Table 2) are found to be sensitive to AIRS cloud fraction (Table. 3 and 4). Winter time series at Thule and Ny-Alesund are shown in Figure 9.

As for MODIS (section 4.1) and SCIAMACHY (section 4.2), AIRS summer TCWV biases are also sensitive to MODIS cloud fraction at Thule (Table 4),

At Sodankyla, no significant correlations are found for the monthly means and the annual cycle, but at inter-annual scale in March with AIRS CF (R > 61 %) and in May with MODIS CF (R = 72).

Most correlations found are sparse temporally and do not show clear features. This might be due to the fact that AIRS TCWV biases are smaller in magnitude (Table. 2) and show a different seasonality compared to MODIS and SCIAMACHY.

## 5    Conclusions

This paper found a general good agreement between satellite TCWV retrievals and coincident measurements from three GPS instruments in the Arctic region. MODIS and SCIAMACHY show overall mean dry biases compared to GPS with some seasonal and latitudinal variation. We generally see better agreement (higher correlation, smaller bias and RMSE) between GNSS and AIRS TCWV time series than between GNSS and MODIS or SCIAMACHY. The seasonal (three-monthly) biases do not exceed 1 kg m$^{-2}$ with AIRS, 2.5 kg m$^{-2}$ with SCIAMACHY(except at Sodankyla for both satellites during summer), and 3.5 kg m$^{-2}$ with MODIS, At Sodankyla, the agreement between GNSS and satellite retrievals is lower for all three satellite measurements. We do not suspect the GNSS data as they passed a selective quality control and outlier detection procedure. Instead, we hypothesize that satellite retrievals are impacted by local effects (cloud cover and canopy).

For MODIS, the inter-annual agreement is getting better with latitude over all seasons except summer. During summer, the inter-annual variability is actually getting worse at higher latitudes sites. These increased summer biases are found to be sensitive to clouds cover. Additionally, MODIS dry biases during some snowy months at Sodankyla are also correlated with cloud fraction.  However, the inaccurate estimation of the surface albedo over a complex mixed surface (snow and nearby canopies) also limits the MODIS retrieval capabilities at Sodankyla.

Summer SCIAMACHY-GNSS TCWV biases are found to be correlated with cloud cover at the higher latitudes sites (Thule and Ny-Alesund), in similar way as MODIS ones, but unlike AIRS. However, Both MODIS and SCIAMACHY seems to be more sensitive to cloud fraction than AIRS as the annual cycle of TCWV bias for both satellites is well correlated with the annual variations of cloud fraction at Thule and Sodankyla,. AIRS time series of TCWV differences to GNSS show a limited link with cloud fraction compared to MODIS and SCIAMACHY (except at Thule) with no clear features. Results reveal anti-correlated monthly differences with AIRS cloud fraction

at Ny-Alesund, probably due to opposite correlation with clouds in winter. Cloud presence is reported to affect satellites TCWV measurements more clearly at Thule compared to both other sites.

Overall, our results suggest a probable link between satellites TCWV biases to GNSS and cloud cover fraction, with contrasted regional and seasonal features. This sensitivity is strong to both AIRS and MODIS cloud fractions at the Thule as both cloud fractions are more correlated at this station, and at all stations during summer. GNSS-AIRS

biases are stronger correlated to the AIRS CF than to MODIS CF, whereas GNSS-MODIS biases are stronger correlated to MODIS CF. The use of two cloud fractions clears out a possible influence of the diurnal differences on studying the cloud impact. This effect is decreasing with latitude, as different sensitivity to both cloud fractions is mostly noticed at Sodankyla, thought to be linked to the diurnal variability.

We suggest that more robust information on clouds is included in the satellite data processing procedures in order to

reduce the TCWV biases in the Arctic, and then improve space-borne instrumental uncertainties. We suggest also using GNSS/TCWV data in the calibration of satellite TCWV measurements.

*Acknowledgements.* This work was developed in the framework of the VEGA project and supported by the French program LEFE/INSU. This work is a contribution to the European COST Action ES1206 GNSS4SWEC (GNSS for

Severe Weather and Climate monitoring; http://www.cost.eu/COST_Actions/essem/ES1206) aiming at the development of the global GNSS network for atmospheric research and climate change monitoring. The authors would like to thank Jan Dousa, GOP, Czech Republic, for providing the reprocessed GPS ZTD data, and the staff from the Climserv data centre at IPSL for providing the ERA-Interim data

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

**Table 1:  Over passing hours of each sensor in universal time (UT) at three GNSS sites**

| Station\instrument | MODIS (UT) | SCIAMACHY (UT) | AIRS(UT) |
|---|---|---|---|
| Sodankyla (67° N,26° E) | 08 – 12  &  17 – 21 | 08 – 11  &  17 – 20 | 09 – 12 &  23 – 03 |
| Thule (76° N,69° W) | 15 – 04 | 16 – 20  &  22 – 02 | 06 – 19 |
| Ny-Alesund (78° N,12° E) | 09 – 22 | 10 – 20 | 23 – 13 |

**Table 2 : Bias, RMSE and linear correlation coefficient between MODIS NIR, SCIAMACHY VIS, AIRS IR clear column TCWV retrievals and GNSS TCWV estimates, at Ny-Alesund (78° N, 12° E), Thule (76° N, 69° W), and Sodankyla (67° N, 26° E). Correlations with significance level > 95% are in bold.**

| | Station (Period) | Season | N of pairs | Bias (kg/m$^2$) | Bias (%) | RMSE (%) | R (%) |
|---|---|---|---|---|---|---|---|
| GNSS vs. MODIS | Ny-Alesund (2001-2014) | Monthly | 168 | 0.4 | 3 | 18 | **96** |
| | | DJF | 13 | 0.4 | 9 | 14 | **77** |
| | | MAM | 14 | 0.0 | -0.6 | 14 | 58 |
| | | JJA | 14 | 0.6 | 4 | 7 | 10 |
| | | SON | 14 | 0.8 | 12 | 13 | **56** |
| | Thule (2004-2014) | Monthly | 132 | 0.6 | 10 | 16 | **98** |
| | | DJF | 10 | 0.3 | 13 | 17 | **83** |
| | | MAM | 11 | 0.4 | 10 | 13 | **71** |
| | | JJA | 11 | 1.1 | 10 | 14 | 15 |
| | | SON | 11 | 0.6 | 13 | 14 | **83** |
| | Sodankyla (2001-2014) | Monthly | 166 | 1.7 | 24 | 33 | **96** |
| | | DJF | 13 | 2.8 | 47 | 48 | 30 |
| | | MAM | 14 | 1.5 | 18 | 19 | **74** |
| | | JJA | 14 | -1.1 | -6 | 9 | 41 |
| | | SON | 14 | 3.5 | 32 | 32 | **76** |
| GNSS vs. SCIAMACHY | Ny-Alesund (2003-2011) | Monthly | 81 | 1.5 | 22 | 27 | **97** |
| | | DJF | - | - | - | - | - |
| | | MAM | 9 | 1.1 | 22 | 23 | **81** |
| | | JJA | 9 | 1.7 | 14 | 14 | **76** |
| | | SON | 9 | 1.9 | 24 | 25 | **76** |
| | Thule (2004-2011) | Monthly | 72 | 0.6 | 6 | 24 | **96** |
| | | DJF | - | - | - | - | - |
| | | MAM | 8 | -0.2 | -5 | 9 | **88** |
| | | JJA | 8 | 1.1 | 10 | 11 | 69 |
| | | SON | 8 | 1.4 | 25 | 26 | **90** |
| | Sodankyla (2003-2011) | Monthly | 98 | 2.4 | 19 | 25 | **90** |
| | | DJF | 8 | 1.1 | 21 | 27 | 26 |
| | | MAM | 9 | 1.4 | 17 | 18 | **71** |
| | | JJA | 9 | 4.9 | 27 | 29 | 19 |
| | | SON | 9 | 1.8 | 16 | 18 | 48 |
| GNSS vs. AIRS | Ny-Alesund (2003-2014) | Monthly | 144 | -0.1 | -8 | 19 | **98** |
| | | DJF | 11 | -0.8 | -22 | 26 | **83** |
| | | MAM | 12 | -0 | -2 | 4 | **97** |
| | | JJA | 12 | 1 | 9 | 9 | **94** |
| | | SON | 12 | -0.6 | -8 | 9 | **96** |
| | Thule (2004-2014) | Monthly | 132 | -0.3 | -18 | 31 | **99** |
| | | DJF | 11 | -0.8 | -41 | 44 | **97** |
| | | MAM | 11 | -0.3 | -9 | 14 | **85** |
| | | JJA | 11 | 0.5 | 4 | 5 | **82** |
| | | SON | 11 | -0.5 | -11 | 12 | **92** |
| | Sodankyla (2003-2014) | Monthly | 142 | 1 | 9 | 14 | **98** |
| | | DJF | 11 | 0.8 | 13 | 17 | 50 |
| | | MAM | 12 | 0.7 | 9 | 9 | **90** |
| | | JJA | 12 | 1.5 | 8 | 10 | **64** |
| | | SON | 12 | 1 | 8 | 11 | **58** |

**Table 3: Correlation coefficients (%) between TCWV biases and coincident cloud cover (AIRS) at Sodankyla (SODA) (67° N, 26° E), Thule (THUL) (76° N, 69° W), and Ny-Alesund (NYAL) (78° N, 12° E) for all months, annual cycle, and inter-annual variability (by season and by month). Correlations with significance level > 95% are in bold.**

| | MODIS | | | SCIAMACHY | | | AIRS | | |
|---|---|---|---|---|---|---|---|---|---|
| | SODA | THUL | NYAL | SODA | THUL | NYAL | SODA | THUL | NYAL |
| Monthly | -3 | **39** | **44** | **29** | **60** | **26** | 12 | **31** | **-42** |
| An-cycle | -38 | **68** | 6 | **75** | **75** | -19 | 36 | 42 | **-94** |
| DJF | 43 | **69** | 53 | -49 | - | - | -18 | 44 | **-63** |
| MAM | 46 | 15 | **58** | -37 | 18 | 5 | 4 | 9 | 17 |
| JJA | 53 | **68** | **58** | 27 | **72** | **72** | 36 | 49 | 56 |
| SON | 14 | **69** | 53 | -42 | -3 | 36 | -24 | 0 | 18 |
| Jan | 18 | 48 | **58** | 30 | - | - | -9 | 18 | -47 |
| Feb | 51 | 52 | 44 | -32 | 47 | 57 | 25 | 20 | 7 |
| Mar | **84** | 17 | **78** | 31 | 32 | 42 | **61** | 21 | 32 |
| Apr | 24 | -10 | 42 | -31 | -26 | 23 | 5 | -18 | 13 |
| May | 43 | 52 | 49 | **-77** | 23 | 30 | 45 | **65** | 34 |
| Jun | 44 | 51 | 0 | 7 | -15 | 34 | -13 | 44 | **-63** |
| Jul | 37 | 57 | **81** | 29 | **75** | **80** | 27 | 29 | 52 |
| Aug | 22 | -32 | **81** | -33 | **73** | **60** | -10 | 16 | -14 |
| Sep | 7 | 2 | **58** | -40 | 7 | 37 | -6 | **-68** | 33 |
| Oct | -12 | -8 | 10 | -29 | 55 | 35 | -24 | 10 | -27 |
| Nov | **71** | **77** | **65** | -27 | - | - | -47 | 16 | -27 |
| Dec | **76** | **70** | **73** | - | - | - | 11 | 34 | -9 |

**Table 4: Correlation coefficients (%) between TCWV biases and coincident cloud cover (MODIS) at Sodankyla (SODA) (67° N, 26° E), Thule (THUL) (76° N, 69° W), and Ny-Alesund (NYAL) (78° N, 12° E) for all months, annual cycle, and inter-annual variability (by season and by month). Correlations with significance level > 95% are in bold. Similar significant correlations to table 3 are shown in red.**

| | MODIS | | | SCIAMACHY | | | AIRS | | |
|---|---|---|---|---|---|---|---|---|---|
| | SODA | THUL | NYAL | SODA | THUL | NYAL | SODA | THUL | NYAL |
| Monthly | **49** | **44** | **19** | -15 | **58** | 16 | -4 | **41** | 4 |
| An-cycle | **70** | **68** | -11 | -19 | **80** | -48 | -46 | 47 | -22 |
| DJF | 31 | 32 | 25 | -23 | - | - | -2 | **67** | 24 |
| MAM | 45 | 39 | 9 | -10 | 6 | -6 | 20 | 7 | 37 |
| JJA | **56** | **64** | 2 | 22 | **69** | -14 | 44 | **63** | 6 |
| SON | 41 | -2 | **58** | -36 | 5 | **64** | -6 | 2 | -12 |
| Jan | 25 | 2 | 0 | 24 | - | - | 4 | 35 | 53 |
| Feb | 52 | 52 | 5 | -12 | 44 | **75** | 21 | **65** | -34 |
| Mar | 46 | 37 | 12 | **73** | 68 | 47 | **70** | 39 | 26 |
| Apr | 49 | 37 | **57** | -15 | -41 | 49 | 32 | 21 | 29 |
| May | **77** | 40 | **66** | **-56** | -4 | **73** | **72** | 37 | -27 |
| Jun | 48 | 19 | 9 | 0 | 34 | 47 | 2 | 18 | **-11** |
| Jul | 24 | **81** | 29 | 34 | 20 | -21 | 13 | **89** | -12 |
| Aug | 39 | **66** | **66** | -18 | 51 | **77** | 39 | 20 | -4 |
| Sep | -15 | 10 | **62** | **-67** | 3 | -55 | -27 | **-54** | 11 |
| Oct | 23 | -5 | 8 | -49 | 42 | 39 | 4 | 26 | 10 |
| Nov | **58** | 21 | 20 | -27 | - | - | -13 | 45 | -46 |
| Dec | 30 | -1 | 43 | - | - | - | 34 | 26 | 4 |

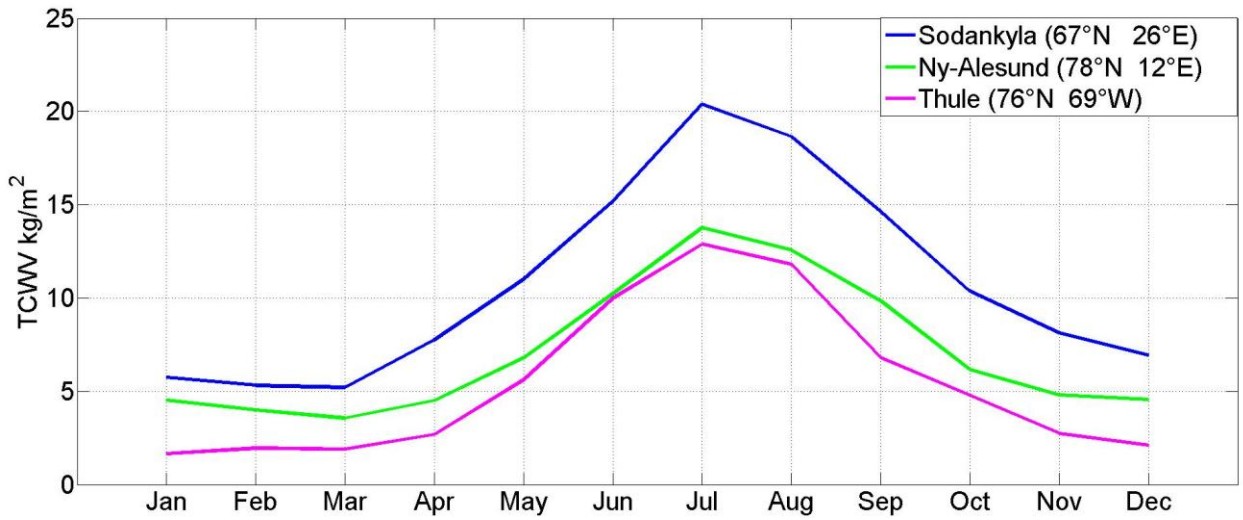

**Figure 1 : Annual cycle of TCWV from GNSS for the period 2004 to 2014 (in kg m$^{-2}$).**

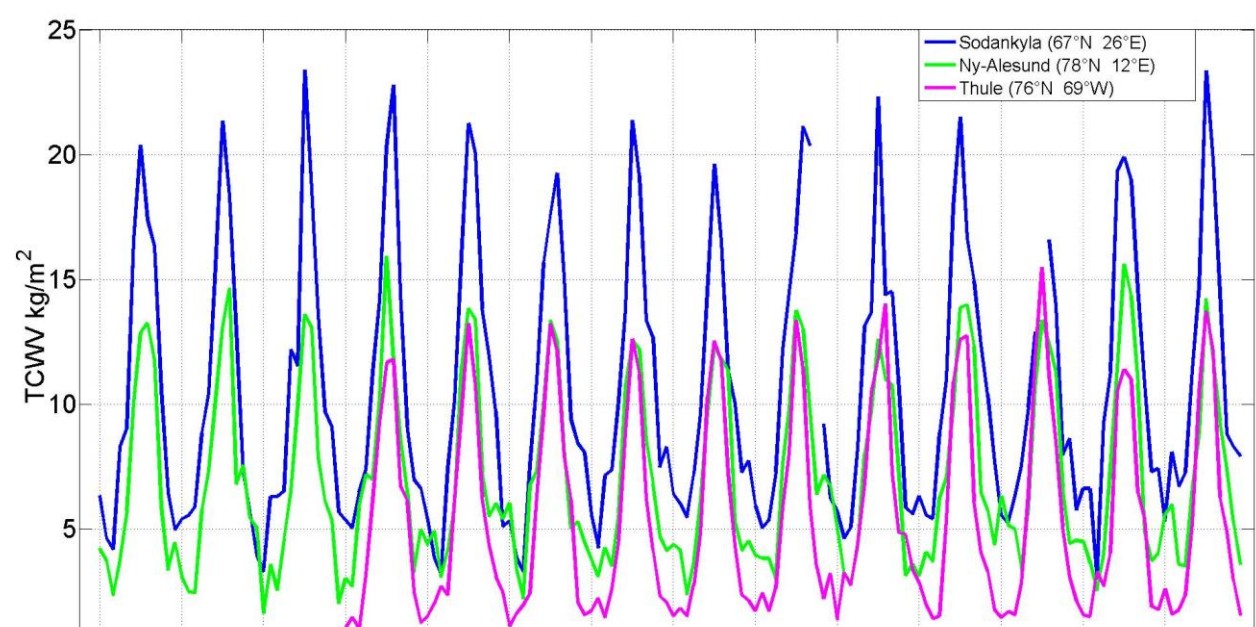

**Figure 2: Monthly time series of TCWV from GNSS over the full period of observation at each site (in kg m$^{-2}$).**

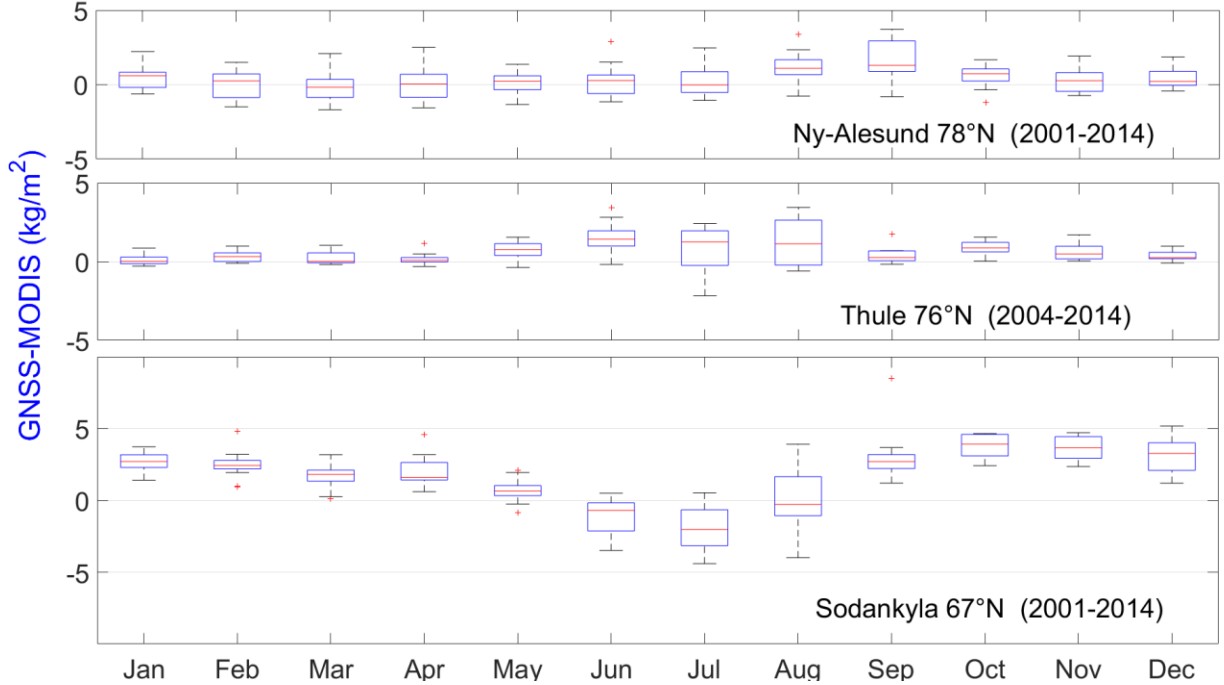

**Figure 3: Box plot of the TCWV differences (GNSS - MODIS) for (2001-2014) at Sodankyla (67° N, 26° E) and Ny-Alesund (78° N, 12° E), and at Thule (76° N, 69° W) for (2004-2014) in kg m⁻². The central red mark indicates the median absolute TCWV difference of the month for the whole period; blue boxes indicate the 25th and 75th percentiles, respectively; black bars (whiskers) extend to ± 1.5 times the inter-quartile range from the median; Outliers are displayed using the '+' symbol.**

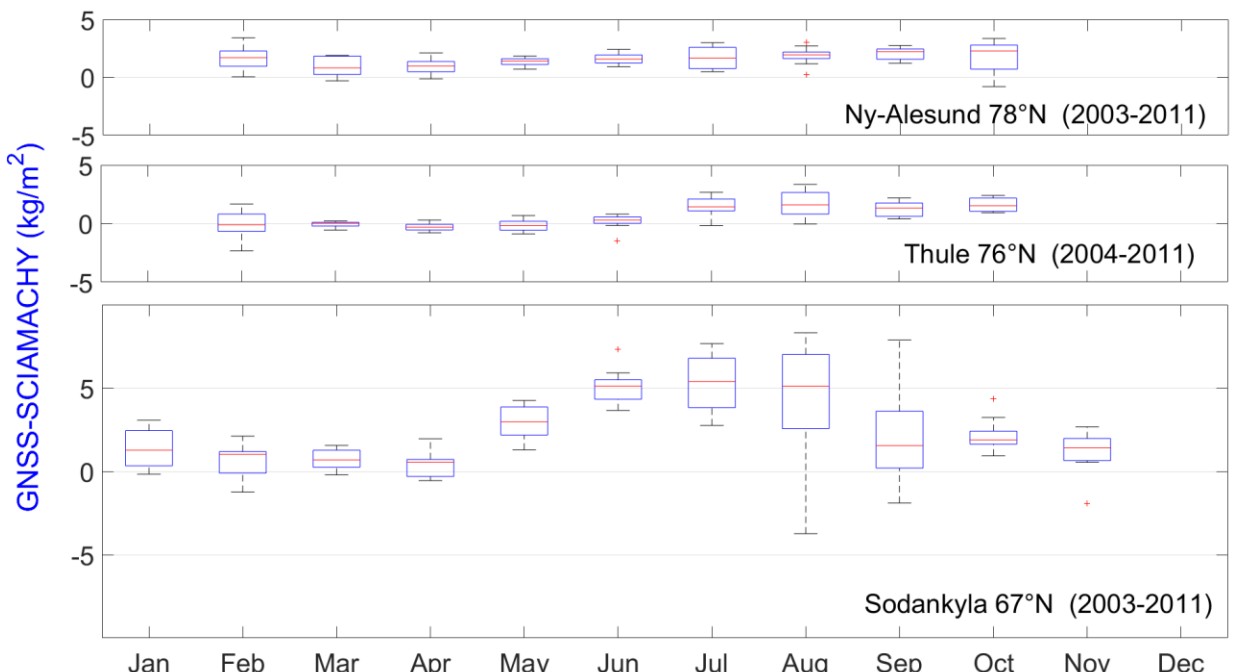

**Figure 4: Box plot of the difference (GNSS - SCIAMACHY) at Sodankyla (67° N, 26° E) and Ny-Alesund (78° N, 12° E) for (2003-2011), and at Thule (76° N, 69° W) for (2004-2011) in kg m⁻². The boxplot indications are same as Fig. 3.**

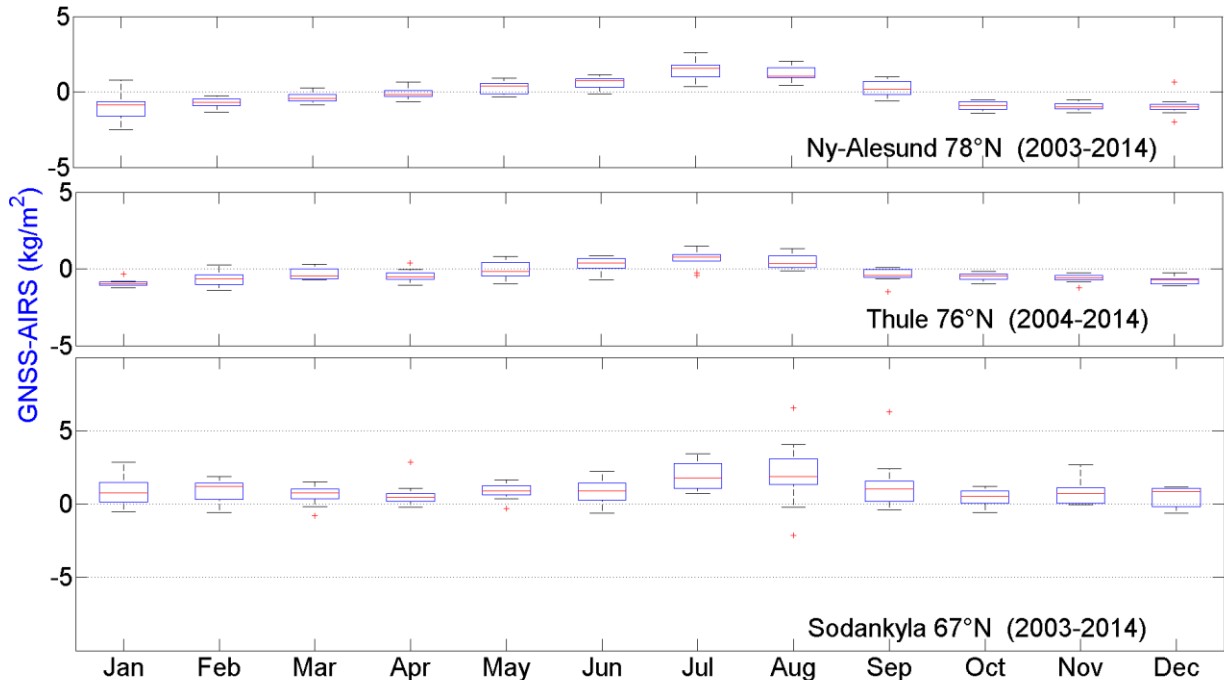

**Figure 5 : Box plot of the difference (GNSS - AIRS) for (2003-2014) at Sodankyla (67° N, 26° E) and Ny-Alesund (78° N, 12° E), and for (2004-2014) at Thule (76° N, 69° W) in kg m⁻². The boxplot indications are same as Fig. 3.**

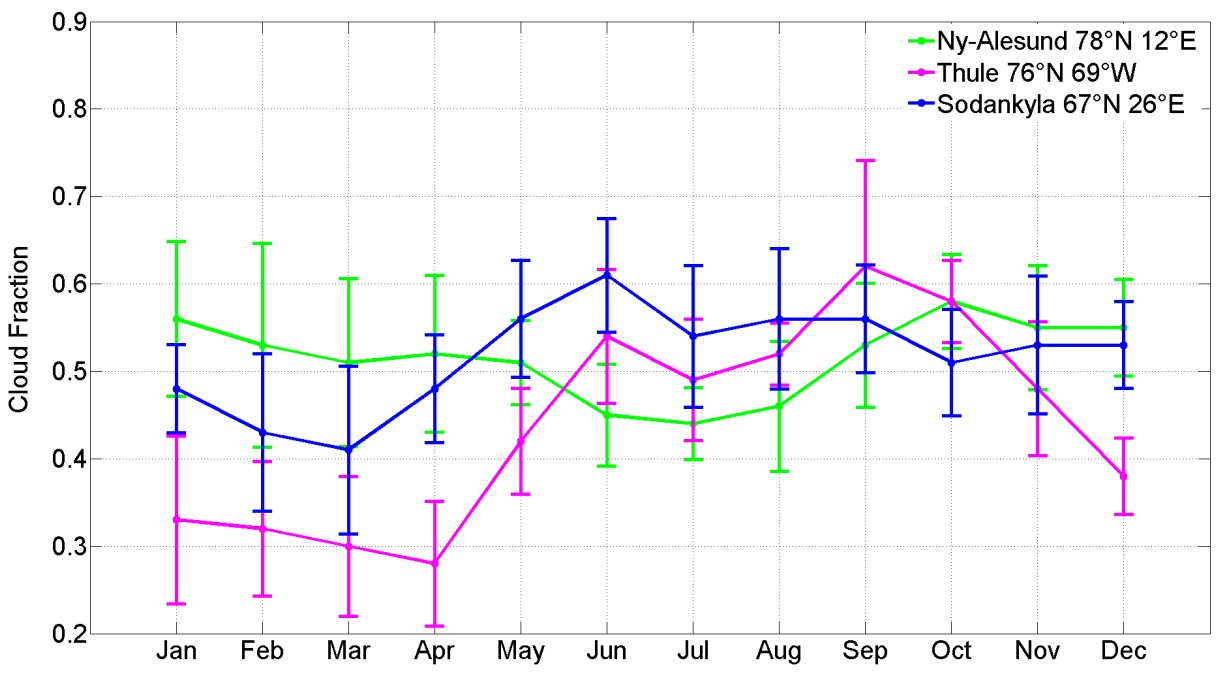

**Figure 6: Annual cycle of AIRS cloud fraction for 2004-2014, the error bars show the standard deviation (1 sigma) of the annual means per month.**

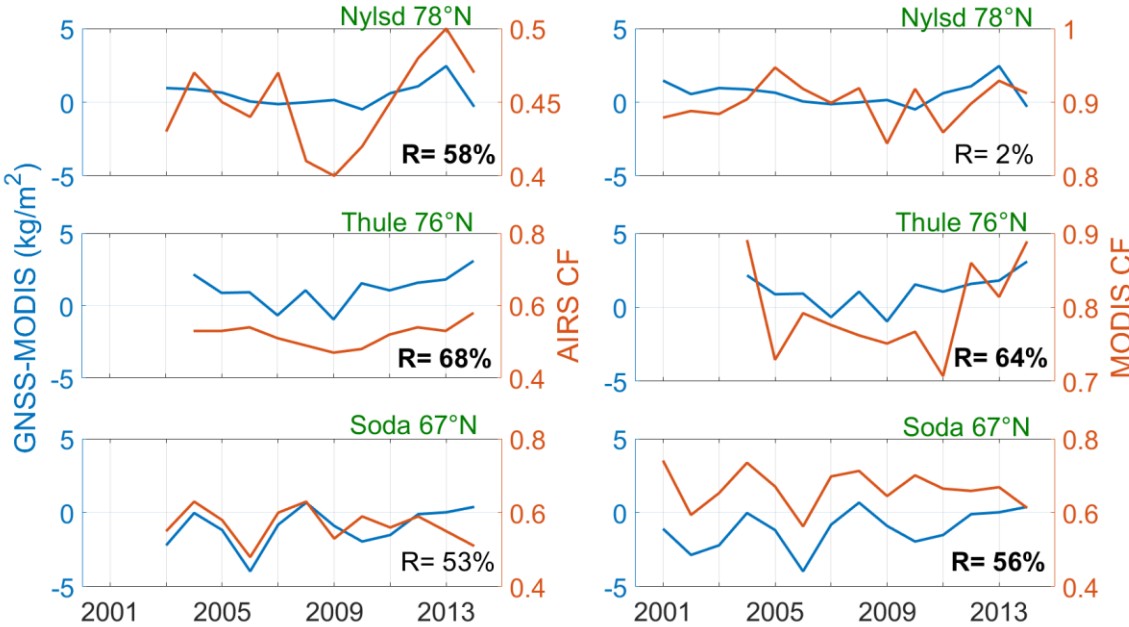

**Figure 7: Summer GNSS – MODIS TCWV differences (kg m⁻²) and AIRS cloud fraction (left side) at Ny-Alesund (78° N, 12° E), Thule (76° N, 69° W) and Sodankyla (67° N, 26° E) from up to down respectively, Right side is the same but with MODIS cloud fraction Significant correlations at the 95% are in Bold.**

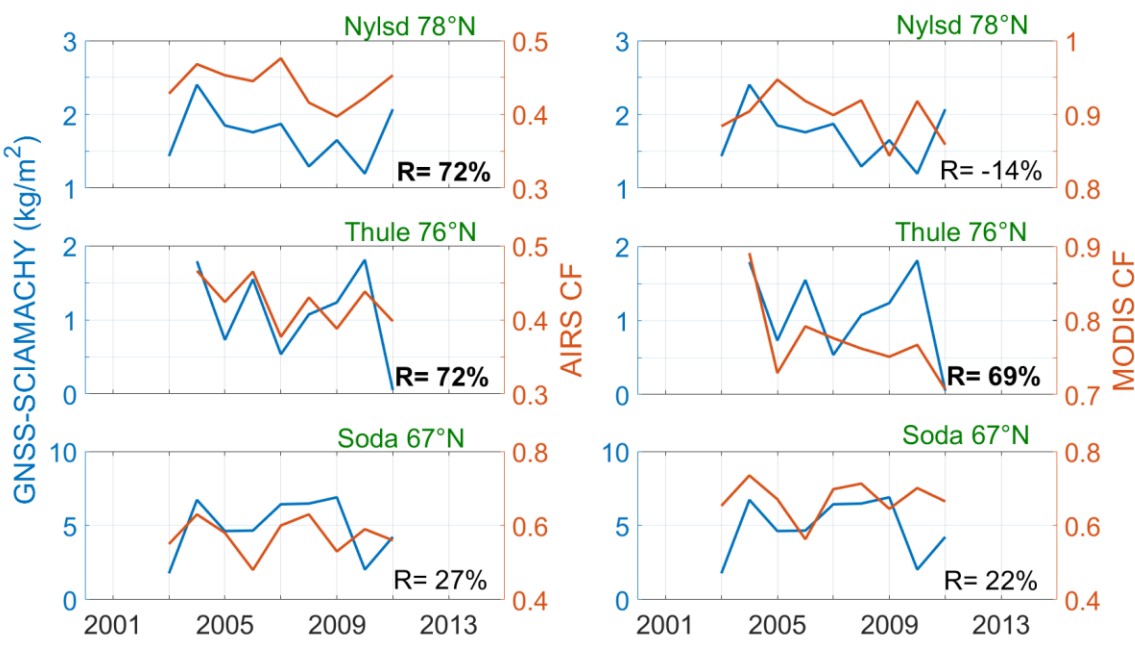

**Figure 8: Summer GNSS – SCIAMACHY TCWV differences (kg m⁻²) and AIRS cloud fraction (left side) at Ny-Alesund (78° N, 12° E), Thule (76° N, 69° W), and Sodankyla (67° N, 26° E) from up to down respectively. Right side is the same but with MODIS cloud fraction. Significant correlations at the 95% are in Bold.**

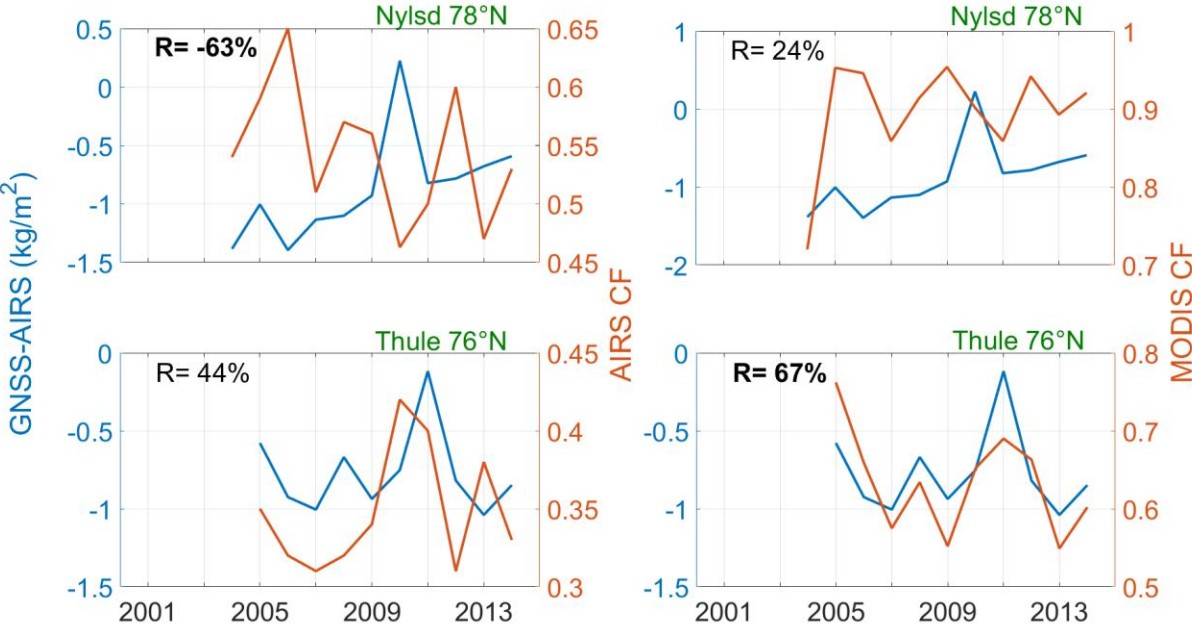

**Figure 9: Winter GNSS – AIRS TCWV differences (kg m$^{-2}$) and AIRS cloud fraction at Ny-Alesund (78° N, 69° W) and**
10   **Thule (76°N, 69°W) from up to down respectively. Right side is the same but for MODIS cloud fraction. Significant**
**correlations are in Bold.**