# Peer review of "Comparison of total water vapour content in the Arctic derived from GNSS, AIRS, MODIS and SCIAMACHY"

_Atmospheric Measurement Techniques, 2017_

## Referee Comment (RC1) · Anonymous Referee #1 · 9 Oct 2017

This article, written about Total Column Water Vapor (TCWV) in the 3 Arctic sites, inter-compares ground-based in situ measurements (GNSS TCWV used as a reference) with TCWV from satellite platforms. The topic and results are attractive for climate research communities for having better overview about instrumental biases, including latitudinal and seasonal variability caused by cloud impact.

The topic is relevant to the scope of AMT. It is not the first in its art, but is targeted to the Arctic and will give valuable information about satellite-derived water vapor accuracy compared to GNSS and cloud impact on the results in Arctic region.

The article relies on known concepts and the authors have used data delivered by

known agencies. Reference GNSS data is processed by Geodetic Observatory Pecny (repro2 solution) with using supporting meteorological data from ERA-Interim, following the best known practices.

The abstract provides a brief overview and a concise summary. The overall structure of the article is clear. The conclusions based on practical data analysis comprise the results of the work with suggestions for better reducing TCWV biases in the Arctic. The article is supported with appropriate references.

The article needs some minor revision before it's ready for publication.

Questions: 1. Can it be explained what does AIRS clear-sky forward mode absolute accuracy 0.2K mean for TCWV derivation (for MODIS given 5-10% TCWV accuracy, page 4, line 35)? 2. Section 4 describes an impact of clouds on satellite TCWV measurements as a source of uncertainties. Is it the only or main factor creating the biases or does there exist any other factors like latitudinal dependence? Is it possible to quantify all the disturbing factors? 3. Figures 7, 8, 9 – is there any idea why Sodankyla is excluded from these figures, however discussed in sections 4.1, 4.2 and 4.3? 4. What can be concluded about the total uncertainties of the space-born instruments and deriving TCWV (instrumental uncertainties, models, . . .)? The outlook for calibrating satellite measurements with GNSS TCWV?

Suggestions: 1. GPS <—-> GNSS, if it cannot be claimed that the authors have used solely GPS-data (i.e. without GLONASS) what is very unlikely for repro2 solution, then the authors should better use GNSS instead of GPS in the title and the following text. 2. Page 4, lines 19-20 as "inter-annual variability (Fig. 2)", and Figure 2: Monthly time series . . . It could be more informative to give inter-annual variability as a table. It is hard to notice/quantify the variability from 10+ year TCWV time series (too much squeezed). Or, it could be pointed on Figures 7, 8 and 9?

Some technical corrections: Typos like Page 10, line 23: "Table. 3" ...

---

## Referee Comment (RC2) · Anonymous Referee #2 · 22 Dec 2017

In this manuscript, authors compare the water vapor product (level 3, monthly means) from several satellite instruments (AIRS, MODIS and SCIAMACHY) against GPS monthly means for over a decade (2004-2014) for three artic stations (Ny-Alesund, Sodankyla, and Thule). Biases and correlations are analyzed at a monthly and seasonal scale. Cloud impact on satellite TCWV observations is also analyzed by studying the correlations between biases and cloud cover. The topic is interesting because there are no such studies in the Artic region, which of great interest because of its special features. The writing is clear, although there are several grammatical errors that must be corrected. I think that the article is appropiate for AMT, since it covers one of the main subject areas of the journal's scope. I have, however, some minor reservations

that should be address before the article is ready for publication.

SUGGESTIONS:

Suggestion 1. I think you should mention the temporal period of study in the abstract.

Suggestion 2. A figure showing time series of cloud cover could be clarifying.

Suggestion 3. Section 2.1 should explain the meaning of "f(lambda, h)".

Suggestion 4. Why do you use only cloud cover from AIRS? This way it is only measured at AIRS passes, and this could influence your results. This should at least be discussed in the paper, or changed to use reanalysis cloud cover or cloud cover from the same satellite (AIRS cloud cover with AIRS TCWV; MODIS cloud cover with MODIS TCWV). Notice for instance that SCIAMACHY removes data with AMF < 0.8 where most cloud scenes are screened out, so it is quite difficult that SCIAMACHY product is affected by cloud cover (except for sampling effects).

Suggestion 5. Section 3.1: You could compare with other regions of the world to see if the positive bias in MODIS is something typical of cold regions or it also happens in other regions. See for example Iberian Peninsula references [1], [2]

Suggestion 6. In several occasions in the results section, you mention possible problems with albedo, specially at Sodankyla. Could you get albedo information (from the satellite products, or from reanalysis) in order to check whether your hypothesis are valid or not?

Suggestion 7. I think you should provide a "theoretical" explanation of the effect that cloud cover should have on the satellite measurements, based on their respective retrieval method. If, for example, clouds are expect to introduce just noise, then you should repeat your calculations of biases vs cloud cover correlations using absolute biases (mean absolute error for example, or the bias without sign). Then you might find more correlations.

Suggestion 8. It would be good and clarifying to have a time series figure of the cloud cover evolution.

Suggestion 9. Page 10, line 31, you mention a correlation that is positive but not significant. I think that if it is not significant, it should not be mentioned. If it is not significant it could be either positive or negative, we cannot say anything about it, no matter that the estimate is positive.

QUESTIONS:

Question 1. Section 2.1: Is there a reason for you to use $0.75°×0.75°$ horizontal resolution? I think Era-Interim products can be downloaded with more resolution (up to $0.125°×0.125°$).

Question 2. Section 2.2: Authors say the product is from Terra platform. Why is not any MODIS Aqua data used?

Question 3. Last paragraph of page 6: it should be more deeply explained. I understand that this is number of cloudy measurements divided by the number of measurements, but what is the limit to consider a cloudy measurement? CF>0? CF>0.05?

Question 4. Section 3.3, Page 8, line 36. wet bias in drier periods and dry bias in moister periods was observed for several satellite instruments in [3], and associated to different spatial resolution (GNSS is local while satellite measurements cover an area of several km). Do you think it could be explained by that reason?

Question 5. From your analysis from Section 4, it does not seem to me that clouds are the only reason behind the satellite TCWV biases. Sure there is some influence, but in the majority of cases the correlations are not significant. So there is probably another factor responsible for the biases.

Question 6. Page 10, line 35, you say "inversely linear". Do you mean linear with negative slope?

TECHNICAL CORRECTIONS:

A. Page 2, line 29: "satellite data" instead of "satellites data".

B. I think you should move Page 2, Line 13 to Section 2.

C. Page 3, Lines 22, 23, use "Zenith", not "Zenithal" (as in Bevis et al. 1992).

D. Page 5, Line 29, Reference without parenthesis.

E. Page 5, line 33: "collocation" instead of "colocation".

F. Page 6, line 9. Specify that 1.30 PM is local time (if it is).

G. Page 6, line 26. "small" instead of "s mall".

H. Page 6, line 29. "versionˆ2 6" I gues something is wrong there.

I. Page 7, Line 12. Write the biases with sign (+0.4), to make clearer that the biase is positive.

J. Page 7, line 25. "bias", not "biases".

K. Page 7, line 33. "pointed" instead of "point".

L. Page 7, line 34. "contain" instead of "contained".

M. Page 11, line 16. Rephrase "is getting better with latitudes".

L. Check references. For instance, page 12, lines 25-30, the reference has several question marks (?) in the authors names.

REFERENCES:

[1] Vaquero-Martínez, J., Antón, M., Ortiz de Galisteo, J. P., Cachorro, V. E., Costa, M. J., Román, R., & Bennouna, Y. S. (2017). Validation of MODIS integrated water vapor product against reference GPS data at the Iberian Peninsula. International Journal of Applied Earth Observation and Geoinformation, 63(July), 214–221.

https://doi.org/10.1016/j.jag.2017.07.008

[2] Bennouna, Y. S., Torres, B., Cachorro, V. E., Ortiz de Galisteo, J. P., & Toledano, C. (2013). The evaluation of the integrated water vapour annual cycle over the Iberian Peninsula from EOS-MODIS against different ground-based techniques. Quarterly Journal of the Royal Meteorological Society, 139(October), 1935–1956. https://doi.org/10.1002/qj.2080

[3] Vaquero-Martínez, J., Antón, M., Ortiz de Galisteo, J. P., Cachorro, V. E., Álvarez-Zapatero, P., Román, R., ... Noël, S. (2017). Inter-comparison of integrated water vapor from satellite instruments using reference GPS data at the Iberian Peninsula. Remote Sensing of Environment. https://doi.org/10.1016/j.rse.2017.09.028
* * *

---

## Author Response (AR1)

The authors would like to thank the anonymous referee for his/her valuable perspectives and suggestions, we are pleased to answer all the questions.

Question 1: Can it be explained what AIRS clear-sky forward mode absolute accuracy 0.2 K means for TCWV derivation (for MODIS given 5-10% TCWV accuracy, page 4, line 35)?

AC Answer 1: The sentence giving the accuracy of AIRS radiances in temperature (Page 6 L21) was replaced with a statement on accuracy of water vapour retrievals:

"The RMSE of the AIRS water vapor profiles is estimated to 10-15% over 2-km layers in the troposphere (Fetzer et al., 2003; Divakarla et al., 2006)."

Question 2: Section 4 describes an impact of clouds on satellite TCWV measurements as a source of uncertainties. Is it the only or main factor creating the biases or does there exist other factors like latitudinal dependence? Is it possible to quantify all the disturbing factors?

AC Answer 2: In this publication, we focus on the clouds effect on satellites measurements despite the use of cloud cleared TCWV products, as one of the main factors affecting the satellites biases, but not the only one. Our answer is resumed in three points:

a. The global validation efforts of the three satellites products are cited in the manuscript. They show the other factors affecting the biases. However, the following text could be added at Page 3 L14 in the introduction:

"The global validation efforts of the used satellites products discussed many factors affecting the satellites biases. For example, both MODIS (Gao and Kaufman, 2003) and AIRS (Fetzer et al., 2006) TCWV retrievals are limited by the accurate initialization of the humidity profile. While SCIAMACHY measurements are independents of initial humidity profile, but affected by other factors like the albedo estimation for different surfaces (Noël, 2007). MODIS measurements are known to be affected by hazy conditions, and to be less accurate over dark surfaces (albedo effect as SCIAMACHY). Generally, satellites measurements are more accurate during clear sky conditions. However cloud clearing is a challenging task. The present publication uses cloud cleared products in order to assess their uncertainties for three Arctic stations. Moreover, it suggests a possible relation between these three satellites biases and the cloud cover making use of an available cloud fraction product to facilitate the study."

b. Results reveal that the inter-annual variability agreement (of MODIS and GPS, or SCIAMACHY and GPS) is getting better with latitude for the Arctic studied stations.
However, the limited extent of the studied area (67°N - 78°N) doesn't really allow discussing the variation of biases with latitude.

c. Overall, we do believe that biases at lower latitude studied site (Sodankyla) are more affected by the surface type of the studied site than by clouds. However, the biases sensitivity to clouds occurrence has more obvious latitudinal feature. This is thought to be linked to the type of clouds dominant within the atmospheric column over both higher latitude studied sites, which is mentioned in the manuscript and suggested to be investigated.

Question 3: Figures 7, 8, 9 – is there any idea why Sodankyla is excluded from these figures, however discussed in sections 4.1, 4.2 and 4.3?

AC Answer 3: Figures 7, 8, and 9 illustrate the correlations only at the two higher latitude stations (Ny-Alesund and Thule) where the impact of clouds is the strongest. The discussion of Sodankyla in Section 4.1, 4.2, and 4.3 is based on the results reported in Table. 3. The lower correlations at Sodankyla are also due to other factors than clouds (vegetated surface and the snow composition in winter as mentioned in the manuscript).

Question 4: What can be concluded about the total uncertainties of the space-born instruments and deriving TCWV (instrumental uncertainties, models...), the outlook for calibrating satellite measurements with GNSS TCWV?

AC Answer 4: We think that the cloud clearing processes is still challenging for MODIS and SCIAMACHY. Additionally, TCWV conversion model of both MODIS and SCIA-MACHY need to be improved to enable more realistic estimation of surface albedo regardless the complexity of the surface (vegetated snow covered surfaces). We can complete the end of the conclusion with:

"...and then improve space-borne instrumental uncertainties. This publication recommends the use of GNSS/TCWV in the calibration of similar satellite measurements."
Suggestions:

1.GPS or GNSS, if it cannot be claimed that the authors have used solely GPS-data (i.e. without GLONASS) what is very unlikely for repro2 solution, then the authors should better use GNSS instead of GPS in the title and the following text.

AC comment: Accepted, GPS will be replaced by GNSS.

2. Page 4, lines 19-20 as "inter-annual variability (Fig. 2)", and Figure 2: Monthly time series. It could be more informative to give inter-annual variability as a table. It is hard to notice/quantify the variability from 10+ year TCWV time series (too much squeezed). Or, it could be pointed on Figures 7, 8 and 9?

AC comment: In this sentence we wanted to highlight that the year to year variations of TCWV at the three stations are smaller than the seasonal cycle (Fig.1). This can mainly be seen for summer values (i.e. the peak values) but not as much for the other seasons. The goal was not to be quantitative in this statement. However, we think that some readers might be interested in additional quantitative assessments. We therefore provide our monthly TCWV data in a supplement. The sentence P4 L19-20 is changed to:

"Figure 2 shows that the year to year variations of TCWV at the three stations are smaller than the seasonal cycle (Fig.1). This can be easily seen for summer values (peak values)."

List of references:

Divakarla, M. G., Barnet, C. D., Goldberg, M. D., McMillin, L. M., Maddy, E., Wolf, W., Zhou, L. and Liu, X.: Validation of Atmospheric Infrared Sounder temperature and water vapor retrievals with matched radiosonde measurements and forecasts, J. Geophys. Res. Atmos., 111(9), 1–20, doi:10.1029/2005JD006116, 2006.

Fetzer, E., Mcmillin, L. M., Tobin, D., Aumann, H. H., Gunson, M. R., Mcmillan, W. W., Hagan, D. E., Hofstadter, M. D., Yoe, J., Whiteman, D. N., Barnes, J. E., Bennartz, R.,
Vömel, H., Walden, V., Newchurch, M., Minnett, P. J., Atlas, R., Schmidlin, F., Olsen, E. T., Goldberg, M. D., Zhou, S., Ding, H., Smith, W. L. and Revercomb, H.: AIRS / AMSU / HSB Validation, , 41(2), 418–431, 2003.

Fetzer, E. J., Lambrigtsen, B. H., Eldering, A., Aumann, H. H. and Chahine, M. T.: Biases in total precipitable water vapor climatologies from Atmospheric Infrared Sounder and Advanced Microwave Scanning Radiometer, J. Geophys. Res. Atmos., 111(9), 2006.

Gao, B.-C. and Kaufman, Y. J.: Water vapor retrievals using Moderate Resolution Imaging Spectroradiometer (MODIS) near-infrared channels, J. Geophys. Res. Atmos., 108(D13), n/a-n/a, doi:10.1029/2002JD003023, 2003.

Noël, S.: Product Specification Document for SCIAMACHY water vapour column swath data derived using the AMC-DOAS method., 2007.

Atmos. Meas. Tech. Discuss., doi:10.5194/amt-2017-195-AC2, 2018 © Author(s) 2018. This work is distributed under the Creative Commons Attribution 4.0 License.
The authors would like to thank the anonymous referee for his/her valuable perspectives and suggestions; we are pleased to discuss the suggestions and answer the questions.

Suggestions

Suggestion 1: I think you should mention the temporal period of study in the abstract.

AC answer: We can add the temporal period of study, but we didn't mention it as it is different for the three sensors. However, a sentence in the abstract is modified

to be (P1 L23) "The comparisons between GPS and satellite data are carried out for three reference Arctic observation sites (Sodankyla, Ny-Alesund and Thule) where long homogeneous GPS time series of more than a decade (2001-2014) are available".

Suggestion 2: A figure showing time series of cloud cover could be clarifying.

AC answer: in the manuscript, we have presented the annual cycle in order to highlight the different seasonality of cloud cover at the three stations (fig. 6). The inter-annual variability is also examined and presented when necessary, station by station (fig. 7, 8, 9). We did not include the monthly time series of cloud cover because it is not easy to see the correlation between cloud cover variations and TCWV biases which are better highlighted in Fig 7, 8, and 9.

Suggestion 3: Section 2.1 should explain the meaning of "f (lambda, h)".

AC answer: In the section 2.1, it is mentioned that ïĄň and H are the latitude and altitude of the station. We completed the sentence with "f (ïĄň, h) accounts for the geographical variation of the mean acceleration due to gravity (Davis et al., 1985)."

Suggestion 4: Why do you use only cloud cover from AIRS? This way it is only measured at AIRS passes, and this could influence your results. This should at least be discussed in the paper, or changed to use reanalysis cloud cover or cloud cover from the same satellite (AIRS cloud cover with AIRS TCWV; MODIS cloud cover with MODIS TCWV). Notice for instance that SCIAMACHY removes data with AMF

more clearly in P6-L36: "AIRS cloud fraction is used for this study as AIRS has longest overpasses (Table.1), which include (partially) both other sensors passing hours over the studied stations."

Suggestion 5: Section 3.1: You could compare with other regions of the world to see if the positive bias in MODIS is something typical of cold regions or it also happens in other regions. See for example Iberian Peninsula references [1], [2]

AC answer: Thank you for providing these references. However, the GPS products are not the same, MODIS spatial sampling is different (L2, L3), and the climate and environmental conditions are too different to compare the results. Actually, our results meet for MODIS all the yearlong at Sodankyla, except winter. Nevertheless, MODIS nearly underestimates GPS all the year at both higher latitudes sites.

Suggestion 6: In several occasions in the results section, you mention possible problems with albedo, specially at Sodankyla. Could you get albedo information (from the satellite products, or from reanalysis) in order to check whether your hypothesis are valid or not?

AC answer: Actually, the albedo hypothesis doesn't concern mainly a direct relationship between the biases and the albedo values, in a similar way to the study of the clouds effect. However, we think that the biases here are linked with misestimating the surface albedo, and that includes underestimating/overestimating. In consequence, a classic approach that correlates simply the albedo and the biases is not expected to usefully help the interpretation.

Suggestion 7: I think you should provide a "theoretical" explanation of the effect that cloud cover should have on the satellite measurements, based on their respective retrieval method. If, for example, clouds are expect to introduce just noise, then you should repeat your calculations of biases vs cloud cover correlations using absolute biases (mean absolute error for example, or the bias without sign). Then you might find more correlations. AMTD
AC answer: Generally, clouds shield parts of the atmosphere (placed under the clouds) to the instrument, so that the observed radiance is only a part of the real one. On the other hand, depending on wavelengths, multiple scattering inside the clouds may even increase the observed radiance. This is handled / corrected by the different retrieval methods in a different way, which may cause both under -or over- estimation of the retrieved TCVW. For example, the air mass correction in the SCIAMACHY data includes a correction for the part of the atmosphere below the cloud, but this relies on some assumptions (e.g. about profile shapes) which might lead to under -or over- correction. In general, clouds are expected to have a systematic effect on the retrieval results (not only noise). This effect is different for the different data sets, which we describe in the manuscript.

Suggestion 9: Page 10, line 31, you mention a correlation that is positive but not significant. I think that if it is not significant, it should not be mentioned. If it is not significant it could be either positive or negative, we cannot say anything about it, no matter that the estimate is positive.

AC answer: Accepted. This sentence has been deleted

**QUESTIONS**

Question 1: Section 2.1: Is there a reason for you to use 0.75\_x0.75\_ horizontal resolution? I think Era-Interim products can be downloaded with more resolution (up to 0.125\_x0.125\_).

AC answer: The ERA-Interim products are archived at IPSL data center at a resolution of 0.75x0.75 degree which is recommended by ECMWF as it is very close to the actual model grid resolution (T255).

Question 2: Section 2.2: Authors say the product is from Terra platform. Why is not any MODIS Aqua data used?

AC answer: Aqua observations of TCWV suffer from many gaps due to interruptions
and downtimes that prevent authors from using it.

Question 3: Last paragraph of page 6: it should be more deeply explained. I understand that this is number of cloudy measurements divided by the number of measurements, but what is the limit to consider a cloudy measurement? CF>0? CF>0.05?

AC answer: Effective cloud fraction meet the condition CF> 0.01, this information was added in P6 L 37 "cloudy measurements are considered for CF>0.01"

Question 4: Section 3.3, Page 8, line 36, wet bias in drier periods and dry bias in moister periods was observed for several satellite instruments in [3], and associated to different spatial resolution (GNSS is local while satellite measurements cover an area of several km). Do you think it could be explained by that reason?

AC answer: The authors don't think that the explanation is that simple, as there were many exceptions to this remark, see section 3.

Question 5: From your analysis from Section 4, it does not seem to me that clouds are the only reason behind the satellite TCWV biases. Sure there is some influence, but in the majority of cases the correlations are not significant. So there is probably another factor responsible for the biases.

AC answer: That's right, clouds have an influence but this effect couldn't be responsible for all the biases, and we refer to this remark in our conclusions.

Question 6: Page 10, line 35, you say "inversely linear". Do you mean linear with negative slope?

AC answer: Yes. The sentence has been clarified.

**TECHNICAL CORRECTIONS:**

AC answer: All accepted.

**Comparison of total water vapour content in the Arctic derived 5 from GPSGNSS, AIRS, MODIS and SCIAMACHY**

Dunya. Alraddawi1, Alain. Sarkissian1, Philippe. Keckhut1, Olivier. Bock2, Stefan. Noël3, Slimane. Bekki1, Abdenour. Irbah1, Mustapha. Meftah1, Chantal. Claud1,4

1OVSQ-LATMOS, University Paris Saclay, Guyancourt 78280, France 2IGN-LAREG, University Paris Diderot, Paris 75013, France 3Institute of Environmental Physics, University of Bremen, 28334 Bremen, Germany

4LMD/IPSL, CNRS, École Polytechnique, Université Paris Saclay, ENS, PSL Research University, Sorbonne

Universités, UPMC Univ Paris 06, Palaiseau, France

15

10

**Corresponding author: Alain Sarkissian (alain.sarkissian@latmos.ipsl.fr)**

20

25

- Abstract. Atmospheric water vapour plays a key role in the Arctic radiation budget, hydrological cycle and hence climate, but its measurement with high accuracy remains an important challenge. Total Column Water Vapour (TCWV) data set derived from ground-based GPS-GNSS\_measurements are used to assess the quality of different existing satellite TCWV datasets, namely from the Moderate Resolution Imaging Spectrometer (MODIS), the Atmospheric Infrared System (AIRS), and the SCanning Imaging Absorption spectroMeter for Atmospheric CHartographY (SCIAMACHY). The comparisons between GNSS and satellite data are carried out for three reference Arctic observation sites (Sodankyla, Ny-Alesund and Thule) where long homogeneous GNSS time series of more than a decade (2001-2014) are available The observation sites (Sodankyla, Ny Alesund and Thule) where long homogeneous GPS time
- available. We select hourly GPS-GNSS data that are coincident with overpasses of the different satellites over the 3 sites and then average them into monthly means that are compared with monthly mean satellite products for different seasons. The agreement between GPS-GNSS and satellite time series is generally within 5% at all sites for most conditions. The weakest correlations are found during summer. Among all the satellite data, AIRS shows
- the best agreement with GPS-GNSS time series, though AIRS TCWV is often slightly too high in drier atmospheres 30 (i.e. high latitude stations during fall and winter). SCIAMACHY TCWV data are generally drier than GPS-GNSS measurements at all the stations during the summer. This study suggests that these biases are associated with cloud cover, especially at Ny-Alesund and Thule. The dry biases of MODIS and SCIAMACHY observations are most pronounced at Sodankyla during the snow season (from October to March). Regarding SCIAMACHY, this bias is
- 35 possibly linked to the fact that the SCIAMACHY TCWV retrieval does not take accurately into account the variations in surface albedo, notably in the presence of snow with a nearby canopy as in Sodankyla. The MODIS bias at Sodankyla is found to be correlated with cloud cover fraction and is also expected to be affected by other atmospheric or surface albedo changes linked for instance to the presence of forests or anthropogenic emissions. Overall, the results point out that a better estimation of seasonally-dependent surface albedo and a better consideration of vertically-resolved cloud cover are recommended if biases in satellite measurements are to be 40

reduced in polar regions.

**Commentaire [DA1]:**

RC2/ Suggestion 1: I think you should mention the temporal period of study in the abstract.

**AC2:**

We can add the temporal period of study, but we didn't mention it as it is different for the three sensors. However, a sentence in the abstract is modified C1 AMTD Interactive comment Printer-friendly version Discussion paper to be (P1 L23)

[revised manuscript text omitted]

**Commentaire [DA2]:**

RC1: Section 4 describes an impact of clouds on satellite TCWV measurements as a source of uncertainties. Is it the only or main factor creating the biases or does there exist other factors like latitudinal dependence? Is it possible to quantify all the disturbing factors?

AC1: The global validation efforts of the three satellites products are cited in the manuscript. They show the other factors affecting the biases. However, the following text could be added at Page 3 L14 in the introduction

(1)

5 where ZTD is the GPS-GNSS\_ZTD estimate, ZHD is computed from the surface pressure (*Davis et al., 1985*):

ZHD =  $0.002277 P_{sfc} / f (\lambda, H)$ ,

Where  $P_{sfc}$  is the surface pressure,  $\lambda$  and H are the latitude and altitude of the station,  $f(\lambda, H)$  accounts for the geographical variation of the mean acceleration due to gravity *[Davis et al., 1985]*. TCWV is converted from the ZWD as:

10 TCWV = ZWD \* K ( $T_m$ ),

15

Where K  $(T_m)$  is a delay to mass conversion factor and  $T_m$  is the weighted mean temperature.

In this study, we used GPS-GNSS ZTD data from the Geodetic Observatory Pecny (Czech Republic) named "repro2 solution" and referred to as GO4 (*Dousa et al., 2017*). This GPS-GNSS solution was produced with a homogeneous and optimized processing strategy. Outliers in the ZTD time series were detected and removed using the range-check and outlier check method described in (*Bock et al., 2014*). ZHD and  $T_m$  were computed from the ERA-Interim reanalysis pressure level data (37 vertical levels between 1000 hPa and 1 hPa, 0.75° x 0.75° horizontal resolution, 6-hourly time resolution) (*Dee et al., 2011*). The data were first interpolated vertically to the height of the GPS-GNSS station and then interpolated horizontally (bi-linear interpolation using the 4 grid-points surrounding the station) to

the location of the station. The 6-hourly  $P_{sfc}$  and  $T_m$  data were then interpolated (with cubic splines) to the times of

- 20
   the GPS\_GNSS\_ZTD data resulting in the final 1-hourly GPS-GNSS\_TCWV dataset.

   In order to overcome the satellite/GPS\_GNSS\_timing error due to limited hours of MODIS/AIRS/SCIAMACHY

   measurements during a month over a fixed point at the surface, the satellites passing hours over the three Arctic GPS

   GNSS\_stations
   were

   defined
   through

   thtp://climserv.ipsl.polytechnique.fr/ixion/index.php).
   For each satellite, only GPS\_GNSS\_TCWV
- 25 corresponding to the over-passes less than 1 hour (Table 1) were used to calculate the corresponding monthly time series.

Seasonal variations of the TCWV over all three sites for a common period of 11 years (2004-2014) exhibit a pronounced seasonal cycle (Fig. 1) with mean values ranging from a maximum in July of 20, 14, 13 kg m-2, to a minimum in winter of 6, 4.5, 2 kg m-2 over Sodankyla, Ny-Alesund and Thule respectively.

- Extreme hourly values could reach 40 kg m-2 (not shown) over Sodankyla. This highest amplitude appears in summer under continental climate conditions. Ny-Alesund and Thule have likely similar seasonal features. However, Thule has drier winter/fall periods due to the Greenland ice sheet climate effect. "Figure 2 shows that the year to year variations of TCWV at the three stations are smaller than the seasonal cycle (Fig.1). This can be easily seen for summer values (peak values). The inter annual variability (Fig. 2) is actually weaker than the seasonal one (Fig. 1).
- 35 2.2 MODIS

The passive imaging spectral radiometer is installed on both platforms (Terra and Aqua) of the Earth Observing System (EOS). Both satellites are launched on polar orbits since 1999 (Terra) and 2002 (Aqua). They overpass the equator at 10:30 a.m. and 1:30 p.m., respectively. The global coverage is provided within 1-2 days, through a nadir-

Commentaire [DA3]: Rc2/ Suggestion 3:

(2)

Section 2.1 should explain the meaning of "f (lambda, h)".

AC 2: In the section 2.1, it is mentioned that  $_{\lambda}$  and H are the latitude and altitude of the station. We completed the sentence with:

**Commentaire [DA4]:**

Page 4, lines 19-20 as "inter-annual variability (Fig. 2)", and Figure 2: Monthly time series. It could be more informative to give inter-annual variability as a table. It is hard to notice/quantify the variability from 10+ year TCWV time series (too much squeezed). Or, it could be pointed on Figures 7, 8 and 9?

AC1: In this sentence we wanted to highlight that the year to year variations of TCWV at the three stations are smaller than the seasonal cycle (Fig.1). This can mainly be seen for summer values (i.e. the peak values) but not as much for the other seasons. The goal was not to be quantitative in this statement. However, we think that some readers might be interested in additional quantitative assessments. We therefore provide our monthly TCWV data in a supplement. The sentence P4 L19-20 is changed to: 5 looking geometry at a solar zenith angle of 45 degrees. The spatial resolution varies between 250 m and 1 km per pixel depending on the spectral band.

[revised manuscript text omitted]

**Commentaire [DA5]:**

RC1: Can it be explained what AIRS clearsky forward mode absolute accuracy 0.2 K means for TCWV derivation (for MODIS given 5-10% TCWV accuracy, page 4, line 35)?

AC1: The sentence giving the accuracy of AIRS radiances in temperature (Page 6 L21) was replaced with a statement on accuracy of water vapour retrievals:. 5 regions are unbiased relative to in-situ radiosondes. Most results indicate a small mean bias that doesn't exceed 10 % with no significant dependency upon cloud amount.

AIRS TCWV data (*Susskind et al., 2014*) used in this study are taken from observations in the ascending orbit mode (version 6, monthly weighted means, level 3) product, namely AIRX3STM\_006. This data set should have high quality retrievals due to the dense orbital coverage at high latitude. Similarly to MODIS data, the 1° by 1° gridded AIRS pixels were screened. The AIRS considered TCWV pixel per station is the same as for MODIS and defined by

 10
 AIRS pixels were screened. The AIRS considered TCWV pixel per station is the same as for MODIS and defined by formula (3). The comparison to GPS-GNSS is done from 2003 to 2014 for Sodankyla and Ny-Alesund and from 2004 to 2014 for Thule according to AIRS and GPS-GNSS data availability.

During this study, we additionally use the AIRS cloud fraction (CF) monthly 1° by 1° data set also in the ascending mode, namely: AIRx3STMv006 from the version 6 (*Kahn et al., 2014*) in order to study possible effects of cloud

15 interference on the satellites observed biases. AIRS cloud fraction is used for this study as AIRS has longest overpasses (Table.1), which include (partially) both other sensors passing hours over the studied stations. AIRS effective Cloud fraction (used here) is computed as the ratio of the number of AIRS cloudy footprints to the total number of AIRS measurements per 1° by 1° cloudy measurements are considered for CF>0.01.

**3 Mean seasonal comparisons and discussion**

**20 3.1 GPS-GNSS vs MODIS**

MODIS time series of monthly means TCWV are compared to monthly means of coincident overpassing (mentioned in Table 1) GPS-GNSS data over Sodankyla and Ny-Alesund for the period 2001-2014, and over Thule for 2004-2014. This difference in the data range is linked to the GPS-GNSS data availability, as GPS-GNSS dataset has some missing values at Thule during 2001-2003. The results show an excellent overall agreement with a high coefficient

- of correlation R > 96 % for the monthly time series (Table 2). High correlation of the monthly time series is indeed expected since the seasonal cycle is very marked at all three sites (Fig. 2). The mean biases are  $\pm 0.4$ ,  $\pm 0.6$ ,  $\pm 1.7$  kg m-2 at Ny-Alesund, Thule, and Sodankyla, respectively (Table 2). The overall positive biases indicate that MODIS generally under-estimates TCWV compared to GPS. This was previously reported over other cold regions of the world, using other versions of GPS-GNSS and MODIS data, for example, over the Tibetan plateau for both stations
- 30 Gaize and Naque (*Liu et al., 2006*). Here we can also notice a latitudinal decrease both in the absolute bias (in kg m-2) and the relative bias, as well as in the root mean square errors (RMSE), which means that the TCWV retrieval is actually more accurate at higher latitudes.

The mean biases and inter-annual variability of the individual months are analysed with boxplots in Fig. 3. A seasonal variation can be seen at all three sites in the bias and in the dispersion (see the inter-quartile range in the

35 boxplots). The largest variations are observed at Sodankyla with large positive biases between September and February, and slightly negative biases between July and August.

Dividing the year into four seasons, the statistics were also calculated and given in Table 2. At Ny-Alesund and Thule the relative bias doesn't exceed 13% regardless of the season and the absolute biases are larger in (June-July-August) JJA and SON (September-October-November). A small wet biases is observed at Ny-Alesund during spring

**Commentaire [DA6]:**

RC2/Suggestion 4: Why do you use only cloud cover from AIRS? This way it is only measured at AIRS passes, and this could influence your results. This should at least be discussed in the paper, or changed to use reanalysis cloud cover or cloud cover from the same satellite (AIRS cloud cover with AIRS TCWV; MODIS cloud cover with AMDIS TCWV). Notice for instance that SCIAMACHY removes data with AMF < 0.8 where most cloud scenes are screened out, so it is quite difficult that SCIAMACHY product is affected by cloud cover (except for sampling effects).

AC 2: The given AIRS cloud cover has to be interpreted as a typical average value only. It is clear that different sensors with different sampling and different retrieval methods have different sensitivity to cloudiness. Cloud cover fraction by AIRS is used as AIRS has the longest overpasses among the three sensors at the three stations (see table 1). So it covers (even partially) the other sensors overpasses. This is mentioned C2 AMTD Interactive comment Printerfriendly version Discussion paper more clearly in P7-L15:

**Commentaire [DA7]: RC2/Ouestion 3:**

Last paragraph of page 6: it should be more deeply explained. I understand that this is number of cloudy measurements divided by the number of measurements, but what is the limit to consider a cloudy measurement? CF>0? CF>0.05?

AC 2: Effective cloud fraction meet the condition CF> 0.01, this information was added in

[revised manuscript text omitted]

**Commentaire [DA8]:** Rc2/Suggestion 9:**

Page 10, line 31, you mention a correlation that is positive but not significant. I think that if it is not significant, it should not be mentioned. If it is not significant it could be either positive or negative, we cannot say anything about it, no matter that the estimate is positive.

AC 2: Accepted. This sentence has been deleted

**Commentaire [DA9]:** RC2/Question 6:

Page 10, line 35, you say "inversely linear". Do you mean linear with negative slope?

AC 2: Yes. The sentence has been clarified.

- 5 Summer SCIAMACHY TCWV biases are found correlated to clouds cover at the higher latitudes sites (Thule and Ny-Alesund), in similar way as MODIS ones, but unlike AIRS. However, SCIAMACHY seems to be more sensitive to cloud fraction than MODIS as the annual cycle of TCWV bias for SCIAMACHY is well correlated with the annual variations of cloud fraction at Thule and Sodankyla, while MODIS annual cycle of biases show this sensitivity to clouds only at Thule. AIRS time series of TCWV differences to GPS-GNSS show a limited link with
- 10

differences with cloud fraction at Ny-Alesund, probably due to opposite correlation with clouds in winter. Overall, our results suggest a probable link between satellites TCWV biases to GPS-GNSS and cloud cover fraction, with contrasted regional and seasonal features. This sensitivity is stronger at the higher latitudes. We suggest that more robust information on clouds is included in the satellite data processing procedures in order to reduce the

cloud fraction compared to MODIS and SCIAMACHY with no clear features. Results reveal anti-correlated monthly

15 TCWV biases in the Arctic, and then improve space-borne instrumental uncertainties. This publication recommends the use of GNSS/TCWV in the calibration of similar satellite measurements.

 Acknowledgements. This work was developed in the framework of the VEGA project and supported by the French program LEFE/INSU. This work is a contribution to the European COST Action ES1206 GNSS4SWEC (GNSS for 20 Severe Weather and Climate monitoring; <a href="http://www.cost.eu/COST\_Actions/essem/ES1206">http://www.cost.eu/COST\_Actions/essem/ES1206</a> aiming at the development of the global GPS-GNSS network for atmospheric research and climate change monitoring. The authors would like to thank Jan Dousa, GOP, Czech Republic, for providing the reprocessed GPS ZTD data, and the staff from the Climserv data centre at IPSL for providing the ERA-Interim data</a>

**Commentaire [DA10]:**

RC1: What can be concluded about the total uncertainties of the space-born instruments and deriving TCWV (instrumental uncertainties, models...), the outlook for calibrating satellite measurements with GNSS TCWV?

AC1: We think that the cloud clearing processes is still challenging for MODIS and SCIAMACHY. Additionally, TCWV conversion model of both MODIS and SCIAMACHY need to be improved to enable more realistic estimation of surface albedo regardless the complexity of the surface (vegetated snow covered surfaces). We can complete the end of the conclusion with:

[revised manuscript text omitted]

|----------------|-----------|---------------------|---------|------------|---------------------------|----------|----------|-----------|
|                |           | Ŧ (                 | Monthly | 168        | 0.4                       | 3        | 18       | 96        |
| SIC            |           | un(
)14          | DJF     | 13         | 0.4                       | 9        | 14       | 77        |
|                |           | les
-2(          | MAM     | 14         | 0.0                       | -0.6     | 14       | 58        |
|                | SIC       | A-/                 | JJA     | 14         | 0.6                       | 4        | 7        | 10        |
|                | IO        | Ny
(2(           | SON     | 14         | 0.8                       | 12       | 13       | 56        |
|                | М         | Ŧ                   | Monthly | 132        | 0.6                       | 10       | 16       | 98        |
|                | /S | ء
012            | DJF     | 10         | 0.3                       | 13       | 17       | 83        |
|                | S         | nule
1-2         | MAM     | 11         | 0.4                       | 10       | 13       | 71        |
|                | NS        | 100                 | JJA     | 11         | 1.1                       | 10       | 14       | 15        |
|                | 9  | (2                  | SON     | 11         | 0.6                       | 13       | 14       | 83        |
|                | 1         |                     | Monthly | 166        | 1.7                       | 24       | 33       | 96        |
|                | đ         | yla
014          | DJF     | 13         | 2.8                       | 47       | 48       | 30        |
|                | 9         | ank
I-2(         | MAM     | 14         | 1.5                       | 18       | 19       | 74        |
|                |           | 300 i               | JJA     | 14         | -1.1                      | -6       | 9        | 41        |
|                |           | S
(2)            | SON     | 14         | 3.5                       | 32       | 32       | 76        |
|                |           |                     |         |            |                           |          |          |           |
|                |           | р (                 | Monthly | 81         | 1.5                       | 22       | 27       | 97        |
|                | 2         | uns
011          | DJF     | -          | -                         | -        | -        | -         |
|                | Η         | Ales
3-2         | MAM     | 9          | 1.1                       | 22       | 23       | 81        |
|                | AC        | Ny-A
(2003       | JJA     | 9          | 1.7                       | 14       | 14       | 76        |
|                | M         |                     | SON     | 9          | 1.9                       | 24       | 25       | 76        |
|                | CIA       | $\widehat{}$        | Monthly | 72         | 0.6                       | 6        | 24       | 96        |
|                | SC        | 'hule
4-2011     | DJF     | -          | -                         | -        | -        | -         |
|                | 2         |                     | MAM     | 8          | -0.2                      | -5       | 9        | 88        |
|                | S         | T 00                | JJA     | 8          | 1.1                       | 10       | 11       | 69        |
|                | NS        | (2                  | SON     | 8          | 1.4                       | 25       | 26       | 90        |
|                | 5         | - ()                | Monthly | 98         | 2.4                       | 19       | 25       | 90        |
|                | 1         | cyla
01          | DJF     | 8          | 1.1                       | 21       | 27       | 26        |
|                | đ         | ank
3-2          | MAM     | 9          | 1.4                       | 17       | 18       | 71        |
|                | 9         | po                  | JJA     | 9          | 4.9                       | 27       | 29       | 19        |
|                |           | s , 0        | SON     | 9          | 1.8                       | 16       | 18       | 48        |
| wGNSS vs. AIRS |           |                     |         |            |                           | -        |          |           |
|                |           | bi ( <del>1</del>   | Monthly | 144        | -0.1                      | -8       | 19       | 98        |
|                |           | sur
01.          | DJF     | 11         | -0.8                      | -22      | 26       | 83        |
|                |           | Iy-Ale
2003-2    | MAM     | 12         | -0                        | -2       | 4        | 97        |
|                | ß         |                     | JJA     | 12         | 1                         | 9        | 9        | 94        |
|                | ΠA        | ZC                  | SON     | 12         | -0.6                      | -8       | 9        | 96        |
|                |           | ħule
4-2014)     | Monthly | 132        | -0.3                      | -18      | 31       | 99        |
|                | N A       |                     | DJF     | 11         | -0.8                      | -41      | 44       | 97        |
|                | SIS       |                     | MAM     | 11         | -0.3                      | -9       | 14       | 85        |
|                | Ð         | T 003               | JJA     | 11         | 0.5                       | 4        | 5        | 82        |
|                | F ₹       | (5                  | SON     | 11         | -0.5                      | -11      | 12       | 92        |
|                | S.        | а <del>(</del>      | Monthly | 142        | 1                         | 9        | 14       | 98 |
|                | 3         | kyl
201-         | DJF     | 11         | 0.8                       | 13       | 17       | 50        |
|                |           | lanl
3-2         | MAM     | 12         | 0.7                       | 9        | 9        | 90        |
|                |           | pos
200          | JJA     | 12         | 1.5                       | 8        | 10       | 64        |
|                |           | Cl                  | SON     | 12         |                           | 8        | 11       | 58        |

Table 2 : Bias, RMSE and linear correlation coefficient between MODIS NIR, SCIAMACHY VIS, AIRS IR clear column TCWV retrievals and GPS-GNSS TCWV estimates, at Ny-Alesund (78° N, 12° E), Thule (76° N, 69° W), and Sodankyla (67° N, 26° E). Correlations with significance level > 95% are in bold.

I

|          | MODIS |      |      | SCIAMACHY |      | AIRS |      |      |      |
|----------|-------|------|------|-----------|------|------|------|------|------|
|          | SODA  | THUL | NYAL | SODA      | THUL | NYAL | SODA | THUL | NYAL |
| Monthly  | -3    | 39   | 44   | 29        | 60   | 26   | 12   | 31   | -42  |
| An-cycle | -38   | 68   | 6    | 75        | 75   | -19  | 36   | 42   | -94  |
| DJF      | 43    | 69   | 53   | -49       | -    | -    | -18  | 44   | -63  |
| MAM      | 46    | 15   | 58   | -37       | 18   | 5    | 4    | 9    | 17   |
| JJA      | 53    | 68   | 58   | 27        | 72   | 72   | 36   | 49   | 56   |
| SON      | 14    | 69   | 53   | -42       | -3   | 36   | -24  | 0    | 18   |
| Jan      | 18    | 48   | 58   | 30        | -    | -    | -9   | 18   | -47  |
| Feb      | 51    | 52   | 44   | -32       | 47   | 57   | 25   | 20   | 7    |
| Mar      | 84    | 17   | 78   | 31        | 32   | 42   | 61   | 21   | 32   |
| Apr      | 24    | -10  | 42   | -31       | -26  | 23   | 5    | -18  | 13   |
| May      | 43    | 52   | 49   | -77       | 23   | 30   | 45   | 65   | 34   |
| Jun      | 44    | 51   | 0    | 7         | -15  | 34   | -13  | 44   | -63  |
| Jul      | 37    | 57   | 81   | 29        | 75   | 80   | 27   | 29   | 52   |
| Aug      | 22    | -32  | 81   | -33       | 73   | 60   | -10  | 16   | -14  |
| Sep      | 7     | 2    | 58   | -40       | 7    | 37   | -6   | -68  | 33   |
| Oct      | -12   | -8   | 10   | -29       | 55   | 35   | -24  | 10   | -27  |
| Nov      | 71    | 77   | 65   | -27       | -    | -    | -47  | 16   | -27  |
| Dec      | 76    | 70   | 73   | -         | -    | -    | 11   | 34   | -9   |

Table 3: Correlation coefficients (%) between TCWV biases and coincident cloud cover at Sodankyla (SODA) ( $67^{\circ}$  N,  $26^{\circ}$  E), Thule (THUL) ( $76^{\circ}$  N,  $69^{\circ}$  W), and Ny-Alesund (NYAL) ( $78^{\circ}$  N,  $12^{\circ}$  E) for all months, annual cycle, and inter-annual variability (by season and by month). Correlations with significance level > 95% are in bold.

---

## Editor Decision (ED1)

**Review of "Comparison of total water vapour content in the Arctic derived from GNSS, AIRS, MODIS and SCHIAMACHY" by Alraddawi et al.**

**General comments**

To my opinion, the authors did not respond satisfactory enough to the comments or questions raised by the two reviewers. Therefore, an additional revision of the manuscript is needed by the authors. In particular:

- Q2 of the first reviewer ("Section 4 describes an impact of clouds on satellite TCWV measurements as a source of uncertainties. Is it the only or main factor creating the biases or does there exist other factors like latitudinal dependence? Is it possible to quantify all the disturbing factors"). I agree with this reviewer that the authors focus almost exclusively on the impact of the clouds on the satellite TCWV measurements. This of course belongs to the scientific freedom of the authors, but (i) the authors should argue then more (in the manuscript) why only this effect is investigated, (ii) the authors should provide a "theoretical" explanation of the effect that cloud cover should have on the satellite measurements, based on their respective retrieval method (=suggestion 7 of second reviewer, which was not taken care of **in the manuscript**), and (iii) they should also first describe the other known impacting factors on the satellite TCWV measurements (like IWV dependence, SZA dependence, seasonal dependence: see the reference Vaquero-Martínez, J., Remote Sensing of Environment (2017), http://dx.doi.org/10.1016/j.rse.2017.09.028, already given by the second reviewer). In this context, I also want to raise the question if the differences you find between Thule and Ny Alesund at one hand and Sodankyla on the other hand cannot be partially ascribed to the differences in latitude (10°) and consequently TCWV amount (this is related to your far too vague statement on page 9, lines 21-22: "So there must be a significantly different sensitivity on the measurements to the atmospheric properties over Sodankyla").
- If you decide to focus only on one aspect (cloud impact) to explain the TCWV GNSS-satellite biases, you should be very convincing. For me, at the moment, it is not. The interpretation relies to a large extent on the correlations calculated between 2 time series of maximum 15 points (see e.g. the graphs in Fig. 7-8). I would therefore ask the authors to do the same analysis with another dataset of cloud cover (e.g. MODIS), as suggested by the second reviewer (suggestion 4). This will make your analysis much more consistent and the interpretation (hopefully) much more convincing. Also, please add the 1-sigma or 2-sigma bars to the annual cycle of the AIRS cloud fraction in Fig. 6, this will give the reader a better idea if the annual cycle is significant or not. By the way, adding such 1-sigma or 2-sigma bars to figures 7 and 8 could also shed another light on the interpretation of those figures. You might also want to add a figure, showing a scatter plot between TCWV GNSS-satellite biases and cloud cover (for the monthly values for example), to illustrate visually the retrieved correlation (coefficient).

- I think you could also draw some firmer conclusions, especially about the TCWV uncertainties achieved for the satellite retrievals (see question 4 of the first reviewer). Please give the numbers of the biases for all the treated satellites (not only for AIRS) , which might be compared here with previous intercomparison studies at high-latitude sites.

**Specific comments**

- Page 2, line 35: please rewrite as "despite the presence of a possible bias in certain specific configuration" and add a reference for this statement here.
- Page 2, line 40: replace with "and found GPS to under-estimate both satellite sensors".
- Page 3, lines 6-7: replace with "uncertainties, accuracies, and limitations of several global sensors/techniques available, which could help imorving the data analysis methods (Bock, 2012; Guerova et al., 2005, 2016)."
- Page 3, line 8: explain the acronym GLONASS
- Page 3, line 15: replace with "The global validation efforts of the used satellite products have pointed to many factors causing satellite biases in TCWV".
- Page 3, lines 17-19: replace with "While SCIAMACHY measurements are independent of the initial humidity profile, they are affected by other factors like the albedo estimation for different surfaces (Noël, 2007). MODIS measurements are known to be affected under hazy conditions…
- Page 4, line 10. Explain that what the weights are in $T_m$ (this is the humidity!).
- Page 6, lines 15-18: replace with "Note that SCIAMACHY data solar dependency results in missing data for winter months. Our study takes place from 2003 to 2011 over Sodankyla and Ny-Alesund and from 2004 to 2011 for Thule."
- Page 7, lines 13-14: please specify what you mean by "longest overpasses". Does it mean that at Thule for instance, AIRS is measuring TCWV continuously between 06-19 UT (see Table 1) (I'm playing the devil's advocate here)?
- Page 7, lines 14-16: replace with "AIRS effective cloud fraction used here is computed as the ratio of the number of AIRS cloudy measurements (CF>0.01) to the total number of AIRS measurements per 1° by 1°.
- Page 8, line 26-27: although I do not doubt your data quality of GNSS and SCIAMACHY, another probably reason for the smaller biases in your study compared to ours (I think you should also only refer to the AMT paper, and not the AMTD paper), is the fact that your study deals only with high-latitude sites, hence sites with lower IWV contents, and hence lower IWV biases between different techniques (as outlined in our AMT paper). To be completely fair, you should compare the values site by site (Ny Alesund, Thule, Sodankyla).
- Page 11, line 26 & 29: replace "don't" by "do not"
- Page 11, line 29: replace impact by impacted
- Page 11, line 31: replace increase by increased
- Page 11, line 35: replace with "Summer SCIAMACHY-GNSS TCWV biases are found to be correlated with could cover at the …."

---

## Author Response (AR2)

The authors would like to thank the editor for his useful suggestions and recommendations.

The according corrections made in the manuscript are highlighted with track changes.

**EC1. Q2 of the first reviewer ("Section 4 describes an impact of clouds on satellite TCWV measurements as a source of uncertainties. Is it the only or main factor creating the biases or does there exist other factors like latitudinal dependence? Is it possible to quantify all the disturbing factors"). I agree with this reviewer that the authors focus almost exclusively on the impact of the clouds on the satellite TCWV measurements. This of course belongs to the scientific freedom of the authors, but (i) the authors should argue then more (in the manuscript) why only this effect is investigated, (ii) the authors should provide a "theoretical" explanation of the effect that cloud cover should have on the satellite measurements, based on their respective retrieval method (=suggestion 7 of second reviewer, which was not taken care of in the manuscript), and (iii) they should also first describe the other known impacting factors on the satellite TCWV measurements (like IWV dependence, SZA dependence, seasonal dependence: see the reference Vaquero-Martínez, J., Remote Sensing of Environment (2017), http://dx.doi.org/10.1016/j.rse.2017.09.028, already given by the second reviewer).**

**AC answer:** We agree with the editor that it is important to mention all the error sources of the same importance as the one that is investigated in more detail. The introduction has been revised accordingly; section 2 reviews some of the known limitations of the satellites retrieval algorithms, and the discussion/Interpretation of results in section 3 is enhanced in this regard.

**In this context, I also want to raise the question if the differences you find between Thule and Ny-Alesund at one hand and Sodankyla on the other hand cannot be partially ascribed to the differences in latitude (10°) and consequently TCWV amount (this is related to your far too vague statement on page 9, lines 21-22: "So there must be a significantly different sensitivity on the measurements to the atmospheric properties over Sodankyla").**

**AC answer:** the larger absolute biases and RMSE at Sodankyla would suggest TCWV dependence, but in fact the relative biases are also larger at Sodankyla, so there are additional error sources in work here.

…………………………………………………………………………………………….

**EC2. If you decide to focus only on one aspect (cloud impact) to explain the TCWV GNSS-satellite biases, you should be very convincing. For me, at the moment, it is not. The interpretation relies to a large extent on the correlations calculated between 2 time series of maximum 15 points (see e.g. the graphs in Fig. 7-8). I would therefore ask the authors to do the same analysis with another dataset of cloud cover (e.g. MODIS), as suggested by the second reviewer (suggestion 4). This will make your analysis much more consistent and the interpretation (hopefully) much more convincing.**

**AC answer:** the study was repeated with MODIS CF data. Results are included in Table 4 and a new Figure, and discussed in section 4.

**Also, please add the 1-sigma or 2-sigma bars to the annual cycle of the AIRS cloud fraction in Fig. 6, this will give the reader a better idea if the annual cycle is significant or not.**

**AC answer:** 1-sigma error bars are added in Figure 6.

**By the way, adding such 1-sigma or 2-sigma bars to figures 7 and 8 could also shed another light on the interpretation of those figures. You might also want to add a figure, showing a scatter plot between TCWV GNSS-satellite biases and cloud cover (for the monthly values for example), to illustrate visually the retrieved correlation (coefficient).**

**AC answer:** the TCWV and CF time series used in this study have been included as supplementary material.

……………………………………………………………………………………………….

**EC3. I think you could also draw some firmer conclusions, especially about the TCWV uncertainties achieved for the satellite retrievals (see question 4 of the first reviewer). Please give the numbers of the biases for all the treated satellites (not only for AIRS), which might be compared here with previous intercomparison studies at high-latitude sites.**

**AC answer:** all satellites absolute biases are added, however it should be noticed that our study uses cloud cleared satellites products unlike previous available studies.

……………………………………………………………………………………………….

**Specific comments**

**Page 3, line 8:** GLONASS is deleted from the sentence as the GOP does include only GPS measurements.

All other comments were addressed as suggested.

…………………………………………………………………………………………………...

[revised manuscript text omitted]

**Commentaire [DA1]:** This part is added to answer the first question of the editor (part iii): "describe the other known impacting factors on the satellite TCWV measurements (like IWV dependence, SZA dependence, seasonal dependence: see the reference Vaquero-Martínez, J., Remote Sensing of Environment (2017),)"

**Commentaire [DA2]:** ZTD used in this study doesn't include GLONASS measurements, it's only GPS as mentioned in 2.1

**Commentaire [DA3]:** This part is added to answer the first question part (i) "the authors should argue then more (in the manuscript) why only cloud effect is investigated"

**Commentaire [DA4]:** This part is added to answer the first question (part ii): "provide a "theoretical" explanation of the effect that cloud cover should have on the satellite measurements, based on their respective retrieval method (=suggestion 7 of second reviewer, which was not taken care of in the manuscript"

**2 Description of the data sets**

**2.1 GNSS**

Originally designed for real-time navigation and positioning, GNSS was rapidly seen as a cheap and accurate technique for measuring TCWV from the ground (Bevis et al., 1992). The principle consists in estimating the propagation delay induced by the atmosphere of the microwave signals emitted by the GNSS satellites and received by ground-based receivers. The Zenith Tropospheric Delay (ZTD) is usually parsed into its wet and hydrostatic components (ZWD and ZHD, respectively for Zenith Wet Delay, and Zenith Hydrostatic Delay). Accurate estimations of surface pressure and a weighted mean temperature are required to convert GNSS ZTD into TCWV using the following formulas (Bevis et al., 1992) :

$$ZWD = ZTD - ZHD, \qquad (1)$$

where ZTD is the GNSS ZTD estimate, ZHD is computed from the surface pressure *(Davis et al., 1985)*:

$$ZHD = 0.002277 \, P_{sfc} / f \, (\lambda, H),$$

Where $P_{sfc}$ is the surface pressure, $\lambda$ and H are the latitude and altitude of the station, f ($\lambda$, H) accounts for the ˘ geographical variation of the mean acceleration due to gravity *(Davis et al., 1985)*.
TCWV is converted from the ZWD as:

$$TCWV = ZWD * K \, (T_m), \qquad (2)$$

Where K ($T_m$) is a delay to mass conversion factor and $T_m$ is the humidity-weighted mean temperature (Bevis et al., 1992).

In this study, we used GNSS ZTD data from the  Geodetic Observatory Pecny (Czech Republic) named "repro2 solution" and referred to as GO4 *(Dousa et al., 2017)*. This ZTD dataset was produced with a homogeneous and optimized processing of GPS observations. Outliers in the ZTD time series were detected and removed using the range-check and outlier check method described in *(Bock et al., 2014)*. ZHD and $T_m$ were computed from the ERA-Interim reanalysis pressure level data (37 vertical levels between 1000 hPa and 1 hPa, 0.75° x 0.75° horizontal resolution, 6-hourly time resolution) (*Dee et al., 2011)*. The data were first interpolated vertically to the height of the GNSS station and then interpolated horizontally (bi-linear interpolation using the 4 grid-points surrounding the station) to the location of the station. The 6-hourly $P_{sfc}$ and $T_m$ data were then interpolated (with cubic splines) to the times of the GNSS ZTD data resulting in the final 1-hourly GNSS TCWV dataset.
In order to overcome the satellite/GNSS timing error due to limited hours of MODIS/AIRS/SCIAMACHY measurements during a month over a fixed point at the surface, the satellites passing hours over the three Arctic GNSS stations were defined through the IXION software (http://climserv.ipsl.polytechnique.fr/ixion/index.php). For each satellite, only GNSS TCWV corresponding to the over-passes less than 1 hour (Table 1) were used to calculate the corresponding monthly time series.

> **Commentaire [DA5]:** Sentence is modified and reference is added to "Explain what the weights are in $T_m$"

> **Commentaire [DA6]:** ZTD used in this study doesn't include GLONASS measurements, it's only GPS

5    Seasonal variations of the TCWV over all three sites for a common period of 11 years (2004-2014) exhibit a pronounced seasonal cycle (Fig. 1) with mean values ranging from a maximum in July of 20, 14, 13 kg m$^{-2}$, to a minimum in winter of 6, 4.5, 2 kg m$^{-2}$ over Sodankyla, Ny-Alesund and Thule respectively.

Extreme hourly values could reach 40 kg m$^{-2}$ (not shown) over Sodankyla. This highest amplitude appears in summer under continental climate conditions. Ny-Alesund and Thule have likely similar seasonal features. However,

10    Thule has drier winter/fall periods due to the Greenland ice sheet climate effect. "Figure 2 shows that the year to year variations of TCWV at the three stations are smaller than the seasonal cycle (Fig.1). This can be easily seen for summer values (peak values)."

**2.2 MODIS**

The Moderate resolution imaging spectroradiometer (MODIS) is installed on both platforms (Terra and Aqua) of the

15    Earth Observing System (EOS). Both satellites are launched on polar orbits since 1999 (Terra) and 2002 (Aqua). They overpass the equator at 10:30 a.m. and 1:30 p.m., respectively. The global coverage is provided within 1-2 days, through a nadir-looking geometry at a solar zenith angle of 45 degrees. The spatial resolution varies between 250 m and 1 km per pixel depending on the spectral band.

MODIS observes the NIR solar radiation reflected by sufficiently bright surfaces and clouds and IR thermal emission

20    in 36 channels covering the spectral region 0.4 - 14.4 μm. It allows the measurement of many other trace gases in addition to clouds and aerosols. In this study we use only the NIR data as they are known to be more accurate.

Five NIR channels are used for retrieving daytime water vapour. They are centred on 0.865, 0.905, 0.936, 0.94, and 1.240 μm, in which all the surface types are sufficiently bright (albedo > 0.1). The extreme channels (0.865 and 1.240 μm) have no water vapour absorption features. They are used to estimate the surface reflectance. The three

25    other channels (0.905, 0.936, and 0.94 μm) absorb water vapour with different sensitivity. The 0.936 μm channel has the strongest absorption sensitivity. TCWV is derived by a differential absorption technique involving channels with absorption and channels without. The accuracy of this product is claimed to be 5–10% *(Gao and Kaufman, 2003)*. Main uncertainties concern the spectral reflectance of surface targets and the uncertainty in the amount of haze for dark surfaces under typical atmospheric conditions *(Gao and Kaufman, 2003)*.

The TCWV data used in this study are from the version 6 of the MODIS instrument on board Terra platform, referenced as "Water vapour near infrared - clear column (bright land and ocean sunglint only): Mean of Daily Mean" *(Gao and Kaufman, 2003; Hubanks et al., 2008)*. The Aqua platform was not used because of many gaps in the measurement. We retrieved global monthly mean files, gridded at 1° by 1°, from the MOD08_M3.006 data

35    stream[1] freely available on:

ftp://ladsweb.nascom.nasa.gov/allData/6/MOD08_M3.

TCWV we extracted from 2001 to 2014 for Sodankyla and Ny-Alesund and from 2004 to 2014 for Thule for the comparison with GPS. The pixel selection method is the following. MODIS data coordinates refer to the centre of

**Commentaire [DA7]:** This part was modified to list and cite known sources of errors in the MODIS TCWV in more details (here instead of the introduction).
* * *
[1] Dataset DOI: http://dx.doi.org/10.5067/MODIS/MOD08_M3.006

5  each gridded pixel, so a single pixel is considered per station (to avoid interpolation and select the nearest pixel to GNSS/IGS stations) and defined as follow:

$$(Lat, Lon)_{(Pixel)} = (lat, lon)_{(station)} + (0.5°, 0.5°),\qquad\qquad(3)$$

Where $(lat, lon)_{station}$ are defined in Table 1 for each of the three stations.

For example the Sodankyla MODIS pixel was selected as follow:

10  $(Lat, Lon)_{(Soda)} = (67°, 26°)_{(table1)} + (0.5°, 0.5°) = (67.5°, 26.5°)$

**MODIS cloud fraction** *(Hubanks et al., 2008; Platnick et al., 2003)* **taken from the same atmospheric product (MOD08_M3.006) is used also to test the sensitivity of the satellite measurements to the presence of clouds. MODIS cloud fraction is defined as the ratio of the count of the lowest two clear sky confidence levels (cloudy**
15  **& probably cloudy) to the total count of scenes per 1° *1°.2.3 SCIAMACHY**

Launched on board the satellite ENVISAT-1in March 2002, the Scanning Imaging Absorption spectrometer for Atmospheric CHartographY (SCIAMACHY) was designed to observe the earthshine radiance and the solar irradiance within limb and nadir alternating viewing geometry. SCIAMACHY nadir and limb observations cover the spectra from Ultra Violet (UV) to NIR (214-2380 nm) at moderate spectral resolution (0.2-1.5 nm). The observed
20  spectra enable the measurement of many other trace gases, as well as clouds and aerosols.

SCIAMACHY can measure water vapour at various wavelengths from the VIS to the SWIR (Short-Wave Infrared). This paper uses TCWV retrieved by the Air Mass Corrected Differential Optical Absorption Spectroscopy method, shortly AMC-DOAS *(Noël et al., 2004)*, where water vapour is measured in nadir mode in the visible part of the spectrum between 688 nm and 700 nm. This method makes use of the similar slant optical depth of both $O_2$ and
25  water vapour to determine an Air Mass correction Factor (AMF) which compensates for insufficient knowledge of the atmospheric and topographic background, like surface elevation and clouds. AMF includes a correction for the part of the atmosphere below the cloud, but this relies on some assumptions (e.g. about profile shapes) which might lead to under- or over-correction of TCWV values.

Though SCIAMACHY TCWV measurements are independent of the initial humidity profile, they are affected by
30  other factors. A dominant error source in SCIAMACHY TCWV retrieval is caused by uncertainties of the atmospheric radiative transfer, mainly due to effects of varying cloud cover and surface albedo for different surfaces *(Wagner et al., 2011)* This error source is estimated to be about 15 % for clear sky observations, and up to 100% in large clouds amounts *(Van Malderen et al., 2014)*. The sensitivity to the surface albedo may cause deviations of up to about 15%, or 6 kg m$^{-2}$ in regions of high surface albedo *(Noël, 2007a; Noël et al., 2004)*. A scatter of about 5 kg
35  m$^{-2}$, caused by atmospheric variability, is usually observed during the inter-comparison with other TCWV datasets *(Noël, 2007b),*

The three stations used in this study were part of the ground-based stations contributing to the SCIAMACHY validation effort *(Piters et al., 2006)* during which water vapour profiles alone were validated over Thule and Sodankyla, while TCWV was additionally validated over Ny-Alesund.
40  TCWV data used in this paper are from *(Noël et al., 2004)*, where all observations with AMF < 0.8 were removed, as well as those performed at solar zenith angles larger than 88°. We apply an extra screening that excludes data with

**Commentaire [DA8]:** This part is added to answer the first question (part ii): "provide a "theoretical" explanation of the effect that cloud cover should have on the satellite measurements, based on their respective retrieval method (=suggestion 7 of second reviewer, which was not taken care of in the manuscript)"

**Commentaire [DA9]:** This part is modified to quantify with citations some known sources of errors in SCIAMACHY TCWV.

[revised manuscript text omitted]

**Commentaire [DA16]:** This table is added in order to present the results with MODIS CF in the same way as those with AIRS CF, which facilitates the comparison.

| | MODIS | | | SCIAMACHY | | | AIRS | | |
|---|---|---|---|---|---|---|---|---|---|
| | SODA | THUL | NYAL | SODA | THUL | NYAL | SODA | THUL | NYAL |
| Monthly | **49** | **44** | **19** | -15 | **58** | 16 | -4 | **41** | 4 |
| An-cycle | **70** | **68** | -11 | -19 | **80** | -48 | -46 | 47 | -22 |
| DJF | 31 | 32 | 25 | -23 | - | - | -2 | **67** | 24 |
| MAM | 45 | 39 | 9 | -10 | 6 | -6 | 20 | 7 | 37 |
| JJA | **56** | **64** | 2 | 22 | **69** | -14 | 44 | **63** | 6 |
| SON | 41 | -2 | **58** | -36 | 5 | **64** | -6 | 2 | -12 |
| Jan | 25 | 2 | 0 | 24 | - | - | 4 | 35 | 53 |
| Feb | 52 | 52 | 5 | -12 | 44 | **75** | 21 | **65** | -34 |
| Mar | 46 | 37 | 12 | **73** | 68 | 47 | **70** | 39 | 26 |
| Apr | 49 | 37 | **57** | -15 | -41 | 49 | 32 | 21 | 29 |
| May | **77** | 40 | **66** | **-56** | -4 | **73** | **72** | 37 | -27 |
| Jun | 48 | 19 | 9 | 0 | 34 | 47 | 2 | 18 | **-11** |
| Jul | 24 | **81** | 29 | 34 | 20 | -21 | 13 | **89** | -12 |
| Aug | 39 | **66** | **66** | -18 | 51 | **77** | 39 | 20 | -4 |
| Sep | -15 | 10 | **62** | **-67** | 3 | -55 | -27 | **-54** | 11 |
| Oct | 23 | -5 | 8 | -49 | 42 | 39 | 4 | 26 | 10 |
| Nov | **58** | 21 | 20 | -27 | - | - | -13 | 45 | -46 |
| Dec | 30 | -1 | 43 | - | - | - | 34 | 26 | 4 |

[Figure]

**Figure 1 : Annual cycle of TCWV from GNSS for the period 2004 to 2014 (in kg m$^{-2}$).**

[Figure]

**Figure 2: Monthly time series of TCWV from GNSS over the full period of observation at each site (in kg m$^{-2}$).**

[Figure]

**Figure 3: Box plot of the TCWV differences (GNSS - MODIS) for (2001-2014) at Sodankyla (67° N, 26° E) and Ny-Alesund (78° N, 12° E), and at Thule (76° N, 69° W) for (2004-2014) in kg m⁻². The central red mark indicates the median absolute TCWV difference of the month for the whole period; blue boxes indicate the 25th and 75th percentiles, respectively; black bars (whiskers) extend to ± 1.5 times the inter-quartile range from the median; Outliers are displayed using the '+' symbol.**

[Figure]

**Figure 4: Box plot of the difference (GNSS - SCIAMACHY) at Sodankyla (67° N, 26° E) and Ny-Alesund (78° N, 12° E) for (2003-2011), and at Thule (76° N, 69° W) for (2004-2011) in kg m⁻². The boxplot indications are same as Fig. 3.**

[Figure]

[Figure]

**Figure 5 : Box plot of the difference (GNSS - AIRS) for (2003-2014) at Sodankyla (67° N, 26° E) and Ny-Alesund (78° N, 12° E), and for (2004-2014) at Thule (76° N, 69° W) in kg m⁻². The boxplot indications are same as Fig. 3.**

[Figure]

**Figure 6: Annual cycle of AIRS cloud fraction for 2004-2014, the error bars show the standard deviation (1 sigma) of the annual means per month.**

**Commentaire [DA17]:** We added the error bar (1 sigma)

[Figure]

**Figure 7: GNSS – MODIS TCWV differences (kg m$^{-2}$) and AIRS cloud fraction at Ny-Alesund (78° N, 12° E) on JJA, and**
**MAM; and at Thule (76° N, 69° W) on JJA, and DJF for (2003-2014).**

[Figure]

**Figure 8: GNSS – SCIAMACHY TCWV differences (kg m$^{-2}$) and AIRS cloud fraction on JJA at Ny-Alesund (78° N, 12°**
**E) and Thule (76° N, 69° W) for (2003-2011).**

[Figure]

**Figure 9: Monthly GNSS-Satellites TCWV differences (kg/m$^2$) and MODIS cloud Fraction at Thule (76°N, 69°W), SCIAMACHY in green, AIRS in blue and MODIS in red.**

**Commentaire [DA18]:** The scatter plot of the satellites biases and the CF as requested (we have chosen Thule as it has the best results)

[Figure]

**Figure 10: GNSS – AIRS TCWV differences (kg m$^{-2}$) and cloud fraction on DJF at Ny-Alesund (78° N, 69° W) and Thule (76°N, 69°W) for (2003-2014).**

---

## Author Response (AR3)

The authors would like to thank the associate editor for his time and valuable remarks, and we invite him to review the new version of the article, hoping that the modifications are more satisfying this time as we tried to address all points.

AE.Q1:  when you describe the annual cycle of the AIRS cloud fraction (Fig 6), you should at least mention if the MODIS cloud fraction has a similar annual cycle at the three sites.

AA.1: Ok, at Thule MODIS CF annual cycle is described as it has a similar annual cycle as AIRS CF, unlike both other stations, this is added in the introduction of chapter 4.

AE.Q2:  thanks for incorporating Table 4! I would certainly keep it. But a figure says sometimes much more than a Table. Therefore, I would also include the MODIS cloud fractions (and the correlations) in Figs. 7, 8, 10. These are the core figures of the paper: with those, you show the reader visually that the cloud fractions might impact the biases. So, please include the MODIS cloud fraction as well, to show what the impact of the selected cloud fraction dataset is on the cloud fraction - GNSS-satellite bias correlations!

AA.2: Ok, figures 7, 8, and 10 were modified to include MODIS CF as well, and biases at Sodankyla are added also to figures 7 and 8 to clarify the differences between sites. Additionally, figure 7 is modified to include only summer time series as the comparison in only one season might present better the differences of different stations.

AE.Q3:  Fig 9 is not so clear: here you do not see any correlation at all, I would say, and you apparently also have problems to interpret it, because you hardly refer to it in the text. So, you can drop it if you want, but anyway, thanks for constructing the figure.

AA.3: Figure 9 is deleted.

AE.Q4:  For the description in Section 4: it is good to start with the obvious findings (most significant correlations in the same months/seasons for both the cloud cover datasets) and then refine to "more or less" common features in both cloud cover datasets and ending by the clear differences. Now, it is a mixture of different elements and it is clearly guided by your first results (with AIRS cloud cover only), which you try to align/strengthen with the MODIS cloud cover correlations. If you have strong scientific arguments to rely more on the AIRS cloud cover dataset in your discussion, you should mention/explain this in the beginning of the section.

AA.4: Section 4 is modified as requested, Please find also additional arguments and references in its introduction.

AE.Q5:  An obvious question to answer in the conclusions too might be if the GNSS-AIRS biases are stronger correlated to the AIRS cloud covers than to the MODIS cloud covers (and same question for the GNSS-MODIS biases).

AA.5: This question is addressed in clearer sentences added to the conclusion.